EMBO
Molecular Medicine

# iPhemap: an atlas of phenotype to genotype relationships of human iPSC models of neurological diseases

Ethan W Hollingsworth[1,2], Jacob E Vaughn[1,2], Josh C Orack[1,2], Chelsea Skinner[1,2], Jamil Khouri[1,2], Sofia B Lizarraga[3], Mark E Hester[4], Fumihiro Watanabe[1], Kenneth S Kosik[5] & Jaime Imitola[1,2,6,*]

## Abstract

Disease modeling with induced pluripotent stem cells (iPSCs) is creating an abundance of phenotypic information that has become difficult to follow and interpret. Here, we report a systematic analysis of research practices and reporting bias in neurological disease models from 93 published articles. We find heterogeneity in current research practices and a reporting bias toward certain diseases. Moreover, we identified 663 CNS cell-derived phenotypes from 243 patients and 214 controls, which varied by mutation type and developmental stage *in vitro*. We clustered these phenotypes into a taxonomy and characterized these phenotype–genotype relationships to generate a phenogenetic map that revealed novel correlations among previously unrelated genes. We also find that alterations in patient-derived molecular profiles associated with cellular phenotypes, and dysregulated genes show predominant expression in brain regions with pathology. Last, we developed the iPS cell phenogenetic map project atlas (iPhemap), an open submission, online database to continually catalog disease phenotypes. Overall, our findings offer new insights into the phenogenetics of iPSC-derived models while our web tool provides a platform for researchers to query and deposit phenotypic information of neurological diseases.

**Keywords** Alzheimer's disease; iPSCs; neurogenetics; neurological diseases; phenogenetics

**Subject Categories** Neuroscience; Stem Cells; Systems Medicine

## Introduction

Deep phenotyping complex human diseases has become increasingly refined by our growing repertoire of molecular, genomic, and computational tools, allowing for precise insights into the specific biology of disease phenotypes and their underlying relationships with genes, or phenogenetics. Despite these recent strides in optimal data acquisition, the functions and phenotypic expression of most genes are not currently known. Animal models have been used for the deep phenotyping of complex neurological diseases (Imitola *et al*, 2004; Sheen *et al*, 2004; Esposito *et al*, 2008); however, there are inherent limitations since animals lack important cells and functions that are present only in humans. The practice of *in vitro* neurological disease modeling with patient-derived cells (Kosik, 2015; Orack *et al*, 2015) has enabled researchers to overcome these evolutionary boundaries, thereby creating a renaissance of human biology *in vitro*.

Obtaining patient-derived cells from induced pluripotent stem cells (iPSCs) has provided the neuroscience field with the prospect of generating cellular models that directly recapitulate the cellular and molecular basis of human disease. In these neurological iPSC models, especially from diseases due to somatic mutations, patient-derived cells are compared to normal controls to detect abnormal cellular characteristics, including differences in proliferation, differentiation, overall cellular integrity, and function, as well as molecular differences. Thus far, hundreds of patient-derived cells have been generated, including neurons, neural stem cells (NSCs), astrocytes, and oligodendrocytes and revealed hundreds of disease phenotypes. However, as this large number of iPSC-derived phenotypes continues to grow, it will be more challenging for researchers in the field to track these phenogenetic relationships. Thus, documenting the state and progress of this field will be increasingly

1 Laboratory for Neural Stem Cells and Functional Neurogenetics, Division of Neuroimmunology and Multiple Sclerosis, The Ohio State University Wexner Medical Center, Columbus, OH, USA
2 Departments of Neurology and Neuroscience, The Ohio State University Wexner Medical Center, Columbus, OH, USA
3 Department of Biological Sciences, University of South Carolina, Columbia, SC, USA
4 Center for Perinatal Research, The Research Institute at Nationwide Children's Hospital, Columbus, OH, USA
5 Department of Molecular Cellular and Developmental Biology, Neuroscience Research Institute, Biomolecular Science and Engineering Program, University of California, Santa Barbara, Santa Barbara, CA, USA
6 The James Comprehensive Cancer Hospital, Columbus, OH, USA
*Corresponding author. Tel: +614 292 0927; E-mail: info@iphemap.org

 

important to reduce redundant work and highlight common, avoidable pitfalls.

Synthesizing this knowledge will also be critical in the search for novel molecular mechanisms of devastating human diseases, especially in sporadic neurological diseases. Moreover, as the development of therapeutic approaches by industry and researchers requires a robust understanding of a targetable pathomechanism, it will be highly advantageous to establish and catalog the overall characteristics of these phenotypes, their degree of interconnectivity, their reproducibility, and the relationship among phenotypes and genetic mutations, especially as drugs that target phenotypes observed across multiple mutations of a disease will be most desirable. Therefore, the clinical and translational utility of these phenotypes will depend upon their characterization, reproducibility, and implementation. Currently, there is no such repository outlining all the reported iPSC-derived neurological disease phenotypes thus far. Hence, a comprehensive database of human CNS cellular phenotypes and an organizing principle of the observed phenotypes and correlated genes are required.

Here, we perform a systematic analysis of current field practices and present a meta-analysis of iPSC-derived CNS cellular phenotypes from neurological diseases. We used manual data mining to extract disease phenotype data of mutant cells from the published studies and combined them to make a field synopsis phenogenetic map. We performed an extensive literature mining and systematic analysis of 663 experimentally observed phenotypes from 71 different gene mutations in 31 adult and pediatric neurological disorders. We synthesized this accrued information into an online knowledge base, The iPS cell phenogenetic map project atlas (iPhemap), that can be searched for curated information on phenotypes found in human iPSC models of neurological diseases and continually updated as new phenotypic information is generated. Our publicly available map of cellular and molecular phenotypes associated with iPSCs demonstrates novel functional relationships among phenotypes and disease-promoting genes. The hundreds of iPSC-derived phenotype–gene relationships catalogued in this study may inform future experimentation to increase reproducibility and rigor of using iPSCs in understanding disease mechanisms and guide high-throughput screening to identify novel compounds to treat neurological diseases.

## Results

### Heterogeneity of methodologies and reporting in neurological disease models with human iPSCs

We included a total of 93 studies out of more than 110 studies initially screened, from which we collected data on phenotypes and genotypes, encompassing 31 neurological diseases that span the pediatric to adult population with a total of 71 gene mutations. We established stringent criteria for the types of studies included in our meta-analysis. These criteria can be found in the flowchart in Fig EV1 and Materials and Methods. The details of the studies are outlined in tabular form in Appendix Table S1 and include 16 categories of pertinent information.

First, we determined that 67% of the studies focused on the investigation of neurodegenerative diseases and within this disease group, we found that diseases with well-characterized somatic mutations were more frequently reported, including Parkinson's disease (29%), amyotrophic lateral sclerosis (ALS) (18%), Huntington's disease (16%), Alzheimer's disease (11%), frontotemporal dementia (FTD) (10%), and spinal muscular atrophy (SMA) (8%), Comparably, we noticed that a large number of the articles modeling neurodevelopmental diseases (30%) studied Rett syndrome (18%) (Fig 1A).

Next, we focused on the reporting of methods employed by each respective study to examine the homogeneity of reporting and field standards. We examined methods that enrich the robustness and reproducibility of a phenotype. For instance, only a small number of studies utilized isogenic lines, which are considered to be a rigorous control where the mutations of patient cells are corrected, $n = 18$, or have submitted gene expression profiles to the Gene Expression Omnibus (GEO) database, $n = 24$, while more than half of the studies used at least more than one control and disease patient line, $n = 49$. Of note, only four studies utilized all four of these methodologies (Fig 1B). Differences in the utilization of these methodologies were observed across all journal types and impact factors, demonstrating that there is no established standard for the reporting of methods nor a defined minimal number of cell lines.

We then analyzed whether there was a common level of similarity in the description of their experimental procedures. Within the 93 studies examined, we identified seven types of categorical methods that were consistently used and we deemed them as the minimal information about iPSC experiments (MiPSCE). Methodology describing how a disease mutation was validated, such as with real-time PCR, showed the most variability among studies; it was reported in 52.7% of studies. Moreover, the inclusion of clinical information of patients, methods describing isolation of primary fibroblasts, or third-party cell repository information was provided in 63.4% of analyzed studies. In contrast, the other five categories were more consistently included, for instance, the methods for gene delivery were reported in all studies, while the details describing cell culture maintenance and procedures of iPSC generation were found in 86 and 87% of studies. Furthermore, the procedures for differentiating iPSCs to other cell types and for performing phenotypic assays were made available in 88 and 96% of studies, respectively (Fig 1C). Complete descriptions of the MiPSCE can be found in the Materials and Methods.

### Taxonomy and clustering of 663 iPS cellular phenotypic traits from human iPSC-derived CNS cell types

Due to the fact that there is striking heterogeneity in the reporting of experiments among the 93 independent studies, it may be argued that based on the lack of uniformity, an analysis of the resulting phenotypes is premature. However, we posit that the collection and analysis of phenotypes culled by curation are required to determine the true state of the field and its limitations. Thus, we performed a comprehensive meta-analysis of phenotypes from iPSC-derived CNS cell types from all 93 studies. First, we extracted all the phenotypes, and in order to maintain the fidelity of what was reported, we used the same semantic description to avoid introducing interpretation bias. The manual curation resulted in 663 distinct cellular phenotypes reported from 71 gene

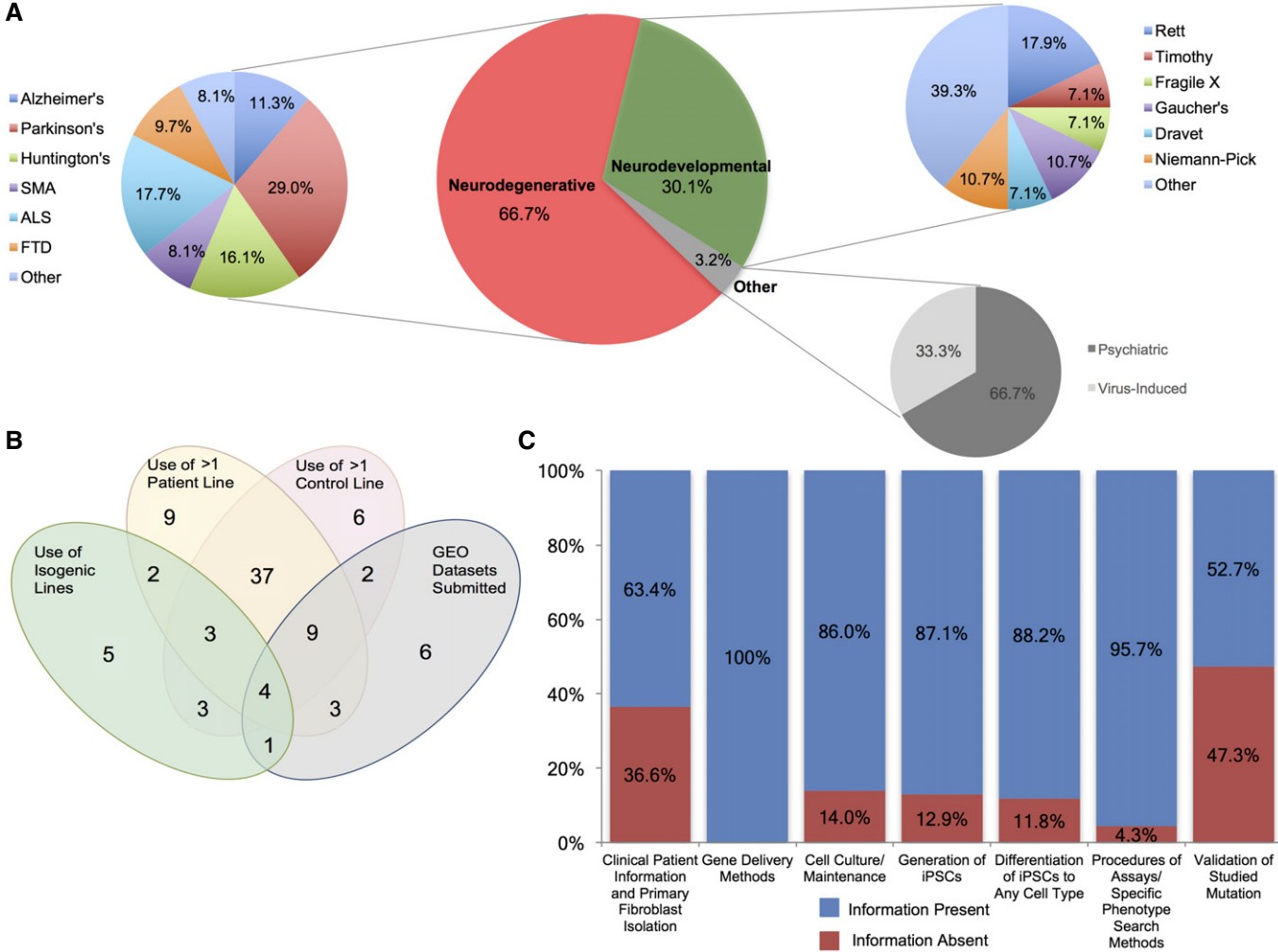

**Figure 1. Diversity of methods utilized to model neurological diseases using patient-derived iPSCs.**

A Pie charts illustrating the percentage distribution of disease groups and specific diseases.

B Venn diagram showing unique and shared experimental techniques among the examined articles.

C Minimal information about iPSC experiments (MiPSCE). Percentage of studies with or without the information for each category is depicted by color. Complete descriptions of the categories can be found in Materials and Methods. SMA, spinal muscular atrophy; ALS, amyotrophic lateral sclerosis; FTD, frontotemporal dementia.

mutations. The goal of iPSC disease modeling is to capture relevant pathological processes by classifying phenotypic traits observed *in vitro*. Therefore, we conceived a taxonomy and grouped all the observed iPSC-derived phenotypes, including the principles and definitions that underlie such classification (Appendix Table S2 and Fig EV2). We used exclusive terminology to group the extracted phenotypic information and defined iPS cellular phenotypic traits (iCPTs), as the distinct phenotypic characteristics of patient-derived cells that are under the genetic influence of the inherited somatic mutation, present in all progeny of the iPSC-derived CNS cell types. These iCPTs were grouped into a set of nine clusters, where each category has its own distinctive definition, comprising all 663 phenotypes, including the following: decreased cellular processes and products (37%), increased cellular processes and products (35%), impairment of expected cellular functions (5%), increased susceptibility to chemical exposure (5%), presence of abnormal

cellular structures (6%), accumulation of molecules (5%), decreased susceptibility to chemical exposure (2%), rescue/recovery from disease phenotypes after chemical treatment (4%), and absence of expected normal phenotypes (1%) (Figs 2A and EV2). Overall, the 663 phenotypes were associated with 42 independent genes, which were mapped by their distinct phenotypic clusters to the human genome (Fig 2B).

## Phenotypic clusters of distinct patient-derived cells

Within neurological diseases, cells other than neurons may be altered, which can be generated *in vitro* to model non-neuronal cellular mechanisms that contribute to neurodegeneration. For instance, in ALS, investigation of patient-derived astrocytes has revealed that the accumulation of abnormal proteins in the mutant astrocytes can be toxic to neurons (Di Giorgio *et al*, 2007). In our

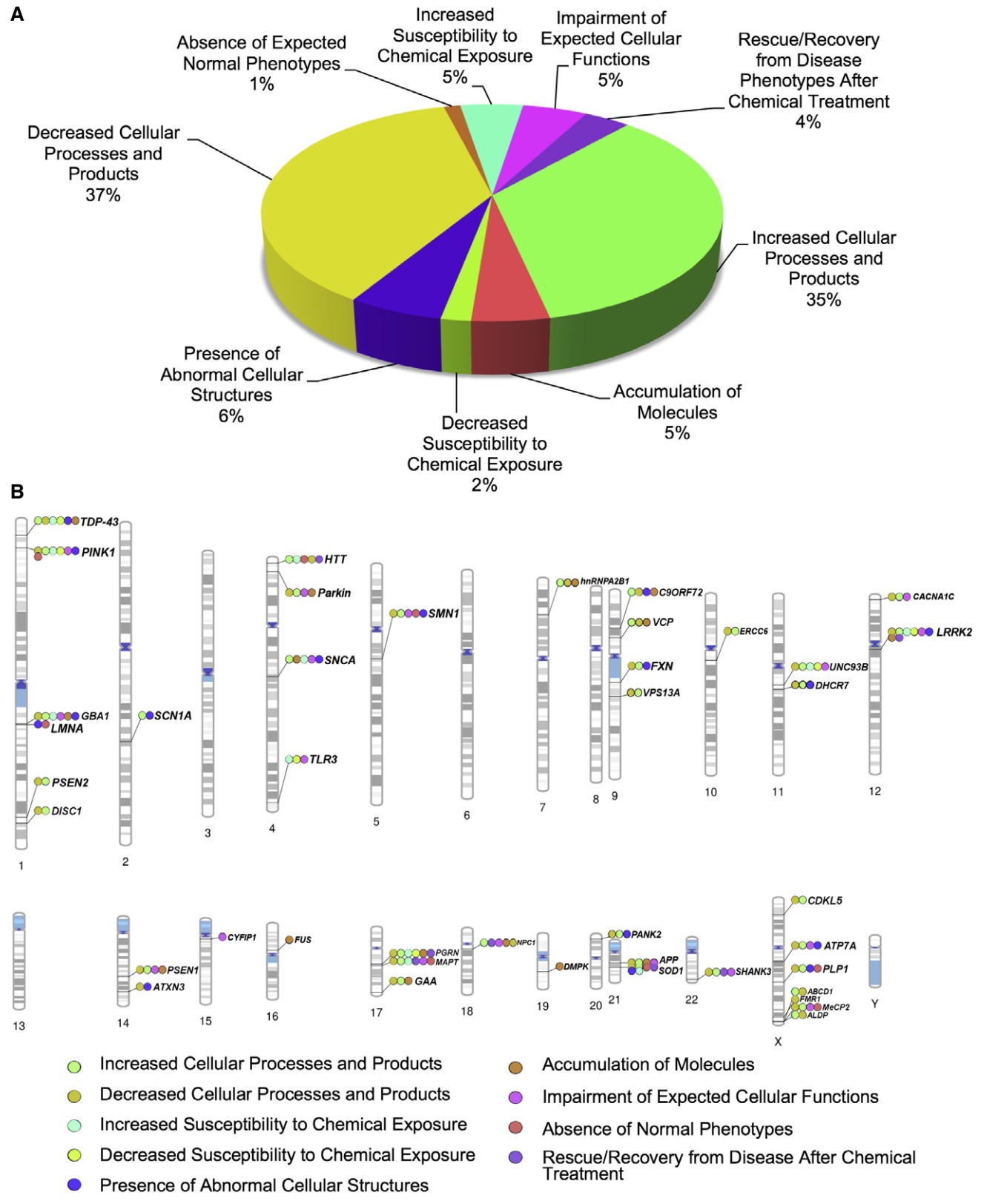

**Figure 2.  Distribution of phenotype clusters and ideogram of iPSC phenotype–genotype associations.**

A   Taxonomy of classes from 663 iPSC-derived cellular phenotypes.

B   Chromosomal maps illustrating the locations of the 42 mappable loci involved in this study. Locus label colors are indicative of the phenotype class observed for each respective locus. These colors were used in the succeeding figures.

analysis, patient-derived cells are comprised of neurons, oligodendrocytes, astrocytes, NSCs, and iPSCs. To gain insight into how the different patient-derived cell types and phenotype clusters are related, we formulated their relationships in a distribution and Circos plot (Fig 3). In comparison with neurons, which exhibit all phenotype classes, the distribution plot revealed that other cells like iPSCs, NSCs, and astrocytes lack the presence of phenotype classes such as "rescue/recovery from disease phenotypes after chemical treatment," observed mostly in neurons. Rather, the phenotype classes, "increased" and "decreased cellular processes and products" comprise the majority of phenotypes reported in the patient-derived CNS cells, excluding oligodendrocytes (Fig 3A and Appendix Table S3). In the Circos plot, which allows for visualization of the distributions of phenotypic clusters within each patient-derived CNS cell type and vice versa, the neuronal phenotype ribbons are greatest on account of the large number of experiments that reported phenotypes observed in patient-derived neurons (Fig 3B). However, given the small number of studies that modeled patient-derived glial cells, the smaller ribbons and absence of certain phenotypic clusters may be an artifact of field practices, opposed to the true state of these diseased cells. Therefore, to normalize our phenotypic findings by cell type, we compared the total number of phenotypes observed in a given cell type to the total number of studies studying that particular cell type. We found that oligodendrocytes had a slightly higher number of phenotypes by studies, followed by neurons (Fig 3C).

## Phenotype and genetic mutation correlates during *in vitro* differentiation of iPSCs

Next, we generated a heatmap displaying the relationships between specific *in vitro* cellular developmental stages of patient-derived cells (i.e., from iPSCs to neurons) and genetic mutations in 31 neurological diseases (Appendix Fig S1 and Table S4). To display the trend of our raw heatmap, we quantified the numbers of phenotypes by the types of diseases and cells included in our analysis (Fig 4A). Notably, we observed a disparity in the emergence of reported disease phenotypes between neurodegenerative and neurodevelopmental disorders. In neurodegenerative disorders like Parkinson's, Alzheimer's, and ALS, phenotypes were chiefly identified at the neuronal stage, with the exception of one iPS cell line with a mutation in *PSEN1* and one line with mutant *FUS* (Fig 4B–F). Indeed, the majority of studies investigated iPSCs compared to neurons, but failed to find phenotypes in Parkinson's disease (PD), Alzheimer's disease (AD), and ALS iPSCs (Nguyen *et al*, 2011; Yagi *et al*, 2011; Liu *et al*, 2012; Sanchez-Danes *et al*, 2012; Chen *et al*, 2014; Muratore *et al*, 2014; Sanders *et al*, 2014; Schondorf *et al*, 2014). The lack of observed phenotypes shows that some, if not all, of the phenotypic outcomes in these diseases occur after differentiation into specific cell types. This phenotypic behavior *in vitro* may model the pathological presentation seen in the human brain, when disease begins in mature neurons and astrocytes that builds up over time. Surprisingly though, this developmental disparity was not present in all neurodegenerative diseases as studies modeling Huntington's detected phenotypes in iPSCs (Jeon *et al*, 2012; Guo *et al*, 2013), which may suggest an early developmental component that is clinically unappreciable (Fig 4G). In contrast, both iPSCs and mature cell types derived from patients with mutations in genes

linked to neurodevelopmental disorders, like *DMPK*, *ERCC6*, and *MECP2*, were altered and exhibited phenotypic abnormalities (Fig 4B–G). Mutations in neurodevelopmental diseases may affect early stages of differentiation, perhaps due to a more pleiotropic role of the mutated gene, leading to a severe dysregulated molecular network, cellular phenotypes, and consequently an early clinical phenotype as seen in Menkes and Pompe disease (Marsden, 2005; Tumer & Moller, 2010).

We also attempted to further analyze if certain mutations show more phenotypes in specific neuronal populations (Hu & Zhang, 2009; Liu *et al*, 2013). We first examined neurons by the type of neurotransmitter they utilize and found that only eleven studies reported such neuronal phenotypes. Despite this limitation, we noted that the majority of phenotypes observed in dopaminergic neurons were from PD-associated mutations, $n = 22$, while GABAergic and glutamatergic phenotypes varied among diseases. Likewise, we gathered that only two studies reported phenotypes in region-specific neurons, including forebrain and midbrain neurons. This analysis, while clearly limited by its small sample size, points to the opportunity for future studies to investigate how particular subpopulations of CNS cell types contribute to disease pathology.

Next, we examined the frequency of phenotypes observed in disease-associated mutations. *Increased oxidative stress in neurons* was the most observed *in vitro* phenotype across different mutations, followed by *accumulation of α-synuclein* and *increased excitability in neurons* (Fig 4H). Conversely, we quantified the number of phenotypes by genes and found that *LRKK2*, $n = 62$, *HTT*, $n = 56$, and *APP*, $n = 53$, exhibited the highest number of distinctive reported phenotypes, followed by *MAPT*, *GBA1*, and *SMN1*. (Fig 4I). However, a caveat of this finding is that these genes with many phenotypes may be a product of being more frequently investigated.

## Network biology analysis of relationships between phenotypes and genes

To further study the association between phenotypes and genotypes, we applied network biology to determine associations between genes and iPSC-derived phenotypes. We computed the degree of association among genes and the collected iCPTs and demonstrated that this network exhibited properties consistent with a scale-free network by fitting a power law to the node degree, topological coefficient, and neighbor connectivity distributions, which exhibited significant *P*-values (Appendix Fig S2). Next, we generated a phenogenetic map based on the network analysis. Phenotypes are color-coded according to their pathological category and numbered based on the 663 distinct phenotypes (Fig 5 and Appendix Table S5). Consistent with the expectation of network biology, a majority of the genes show few phenotypes, but a few genes behave as "hubs" showing a high degree of connectivity to many phenotypes. In addition, multiple phenotypes connect distinct hubs of genes that may belong to the same or different diseases (Fig 5). Other gene and phenotype sub-networks are not connected and remain separate from the highly overlapping network. The complete map is depicted in Appendix Fig S3.

In this analysis, we found 73 phenotypes expressed across 27 different gene loci (Fig 5). These genetic associations are based upon the similar occurrence of observed phenotypes in more than

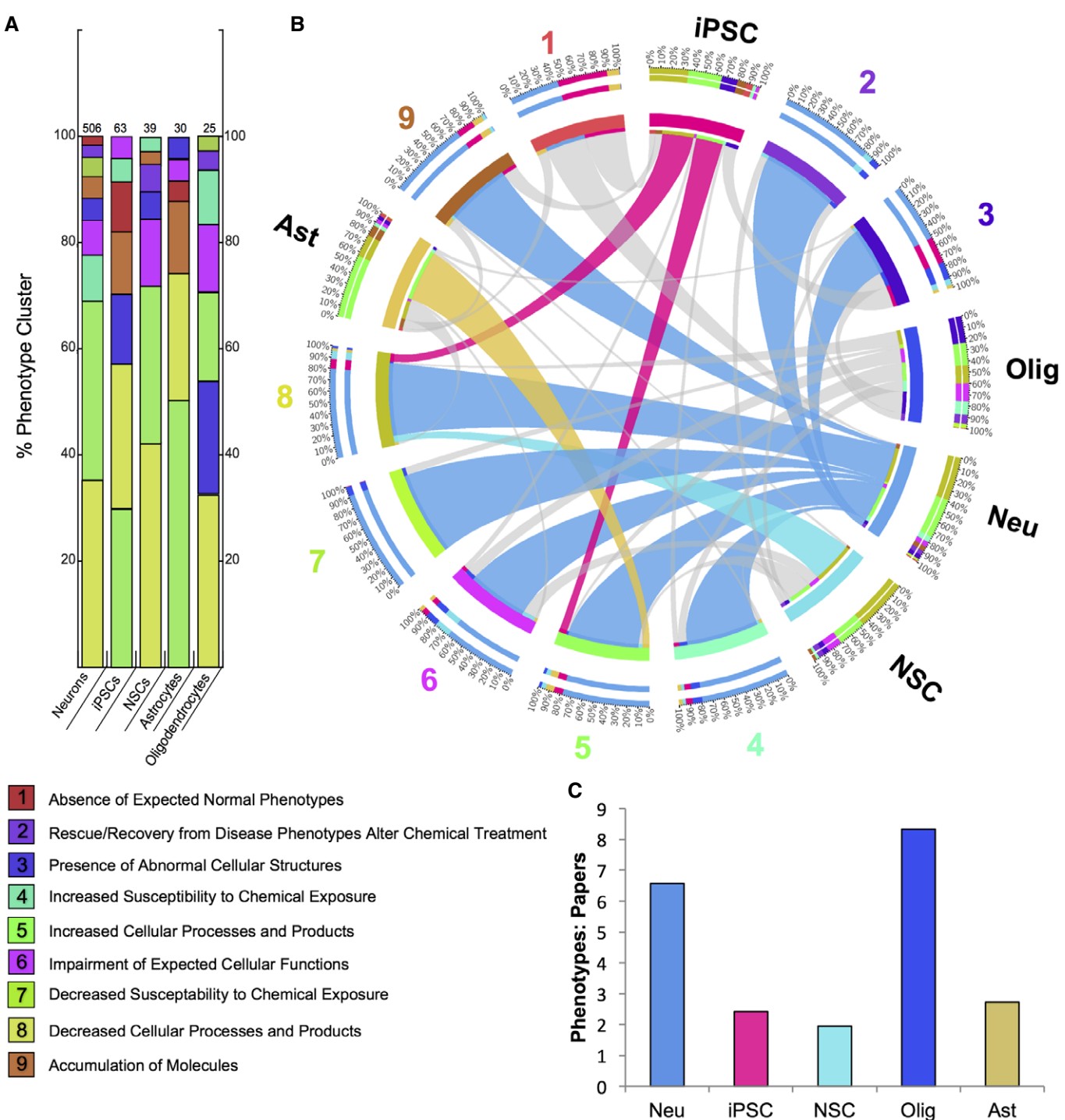

**Figure 3. Phenotypic classes by patient-derived cell type from 663 annotated phenotypes and phenotype: paper metric.**

A  Distribution of the phenotype classes within each CNS cell type with total number of phenotypes listed above each respective column.

B  Circos plot of phenotype classes by cell type and vice versa is depicted by connecting ribbons, with the width of each band proportional to the percent composition and the top-most ribbons highlighted. The neuronal ribbons (blue) were found to connect to and be largest for almost every phenotypic class. The outer track indicates the numeric percentage of phenotypic classes comprising each cell type.

C  Metric of total phenotypes per cell type with respect to the total number of studies that investigated that particular cell type.

one gene, and we refer to them as *overlapping phenotypes*. We observed 55 overlapping phenotypes between genes in neurons alone (Appendix Table S6). Several overlapping neuronal phenotypes were singular to a disease, such as *an increase in α-synuclein*, which was shared by all Parkinson mutations, while *increased oxidative stress*, in contrast, overlapped with multiple

diseases and mutations. (Appendix Table S6). The phenotypes spanning multiple diseases revealed novel associations, such as *decreased neurite outgrowth in neurons*, associated with mutations in *HTT, SMN1,* and *LRRK2,* which have not been related previously. Another new association was *increased excitability in neurons,* correlating with disease-causing mutations in *C9ORF72, SCN1A, TDP-43,* and *SMN1.* (Appendix Table S6). Overlapping phenotypes were also found in other patient-derived cell types. For instance, astrocytes and NSCs were found to show correlation between *intranuclear RNA foci* in cells carrying genetic defects in *DMPK* and *C9ORF72* (Appendix Tables S7 and S8). In oligodendrocytes, the overlapping phenotypes were metabolic alterations associated with Leukodystrophy mutations (Appendix Table S9). Notably, no overlapping phenotypes were seen in iPSCs.

We also studied phenotypes that were most associated with gene mutations responsible for a specific disease or *concordant phenotypes*. For example, in Alzheimer's disease, we noted that all associated genes (*APP, PSEN1,* and *PSEN2*) were reported to show an *increase in Aβ in neurons* (Fig EV3A). In addition, we detected one AD-linked gene, *APP,* to be most concordant with an AD cell line derived from a sporadic-diseased patient with no known mutation, or "*Sporadic*" in Fig EV3A and Appendix Table S10, the only sporadic line included in our analysis of iPSCs with somatic mutations. The two genotypes show seventeen phenotypes spanning multiple cell types, such as *increased levels of ER stress in astrocytes* and *increased levels of binding protein (BiP) in neurons,* suggesting that even sporadic disease may share phenotypic alterations with the genetic-driven disease. Moreover, we observed that some genes, such as *LRRK2* and *PINK1,* both linked to PD, share more concordant phenotypes, $n = 10$, than other PD-associated genes (Fig EV3B). The phenogenetic network not only provides new information, but solidifies previously established genetic associations based on disease types, as shown by the multiple connections between the *PINK1* and *LRRK2* loci (Figs 5 and EV3, and Appendix Fig S3).

## Phenotype and gene ontology comparison

Gene ontology is defined as the functional annotation of phenotypes from individual genes that help to determine their function (Ashburner *et al*, 2000). We investigated the novelty of our phenotype–gene associations and observed a number of phenotypes already established in gene ontology databases. For example, *TDP inclusions* ($P = 9.33 \times 10^{-14}$) in our study corresponded to *formation of cytoplasmic inclusions* ($P = 6.02 \times 10^{-4}$) in gene ontology (Table 1 and Fig EV4). However, there are a significant number of

associations, $n = 15$, that were novel, suggesting that our analysis is expanding the phenotype ontology pool for these human mutations, which could lead to an enrichment of the gene ontology for these genes in the context of human iPS cell models.

## Phenogenetic relationship between molecular and cellular phenotypes in patient-derived cells

We were interested in establishing relationships between molecular and cellular phenotypes and determining if the *in vitro* developmental phenotypic disparity between neurodegenerative and neurodevelopmental disorders would be preserved at the molecular level, since altered gene expression may be the substrate for cellular alterations. Although the purpose of this analysis was not to imply causality, this correlation is nonetheless important to demonstrate how molecular phenotypes can be used as a tool to inform future cellular phenotype assays, especially considering that analysis of cellular phenotypes may be technically challenging and impacted by experimental noise. We made use of the GEO where studies deposited transcriptome data. The analysis was limited by the small number of studies that had published expression data, $n = 24$, and from these, only 10 studies fulfilled our inclusion criteria. The full details of our criteria can be found in the Materials and Methods section. Briefly, it is required that such an analysis was not published in the original study and the expression data from at least three patient-derived and three control cells were available (Appendix Table S1). We then performed the functional analysis of molecular profiles and displayed the data in treemaps.

First, we examined iPSCs containing mutations linked to neurodegenerative disorders. For example, iPSCs containing *LRRK2* mutations show some minor abnormalities in their gene expression profile as we documented *STRN3,* a gene involved in dopamine receptor signaling, to be most upregulated (Osterhout *et al*, 2015). These subtle changes correlated with the absence of reported cellular phenotypes (Appendix Fig S4B). Similarly, iPSCs with *SNCA* mutations show slight downregulation of genes and of molecular pathways, like dopamine signaling, but lacked any reported cellular phenotypes (Appendix Figs S4C and D, and S5A and B). These analyses reveal minor alterations in genes and pathways in cells without observed cellular phenotypes.

In contrast to the PD-linked genes, iPSCs derived from patients with *FXN, HTT,* and *ERCC6* mutations were significantly altered at both the molecular and cellular levels (Appendix Figs S4E–J and S5C–D). For instance, iPSCs derived from patients with *ERCC6* mutations show many changes to their gene

---

**Figure 4.  Quantification of phenotypes by genes and developmental stage.**

A    Schematic diagram depicting developmental timeline of iPSC-derived cells included in analysis.

B–F  Percent distribution plots of (B) iPSC, (C) NSC, (D) astrocyte, (E) oligodendrocyte, and (F) neuronal phenotypes reported for genes linked to neurodegenerative, neurodevelopmental, or other (psychiatric and viral-induced) disorders. Each data point represents a specific disease. One-way analysis of variance (ANOVA) with Bonferroni multiple comparisons tests was performed (NDeg, $n = 13$; NDev, $n = 15$; Other, $n = 3$). Data are expressed as mean percentage $\pm$ s.e.m., *$P < 0.05$, **$P < 0.01$.

G    Distribution of phenotypes by pluripotent, progenitor, and postmitotic cell type for Alzheimer's disease (AD), Parkinson's (PD), Huntington's disease (HD), and Rett syndrome. Two-way ANOVA with Tukey's multiple comparisons test was performed. *$P < 0.05$.

H    Quantification of number of genes observed by phenotype.

I    Quantification of observed phenotypes per gene.

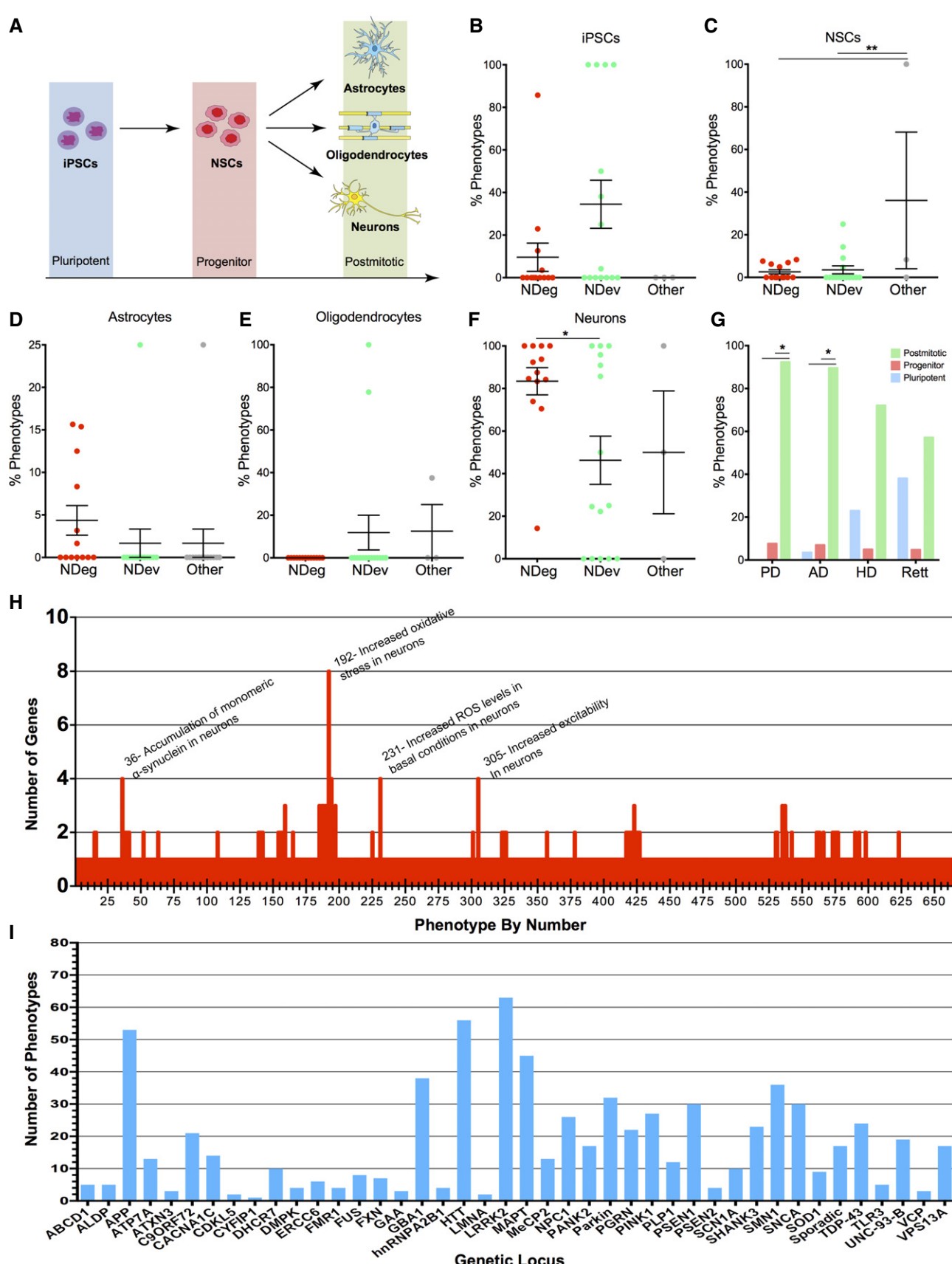

**Figure 4.**

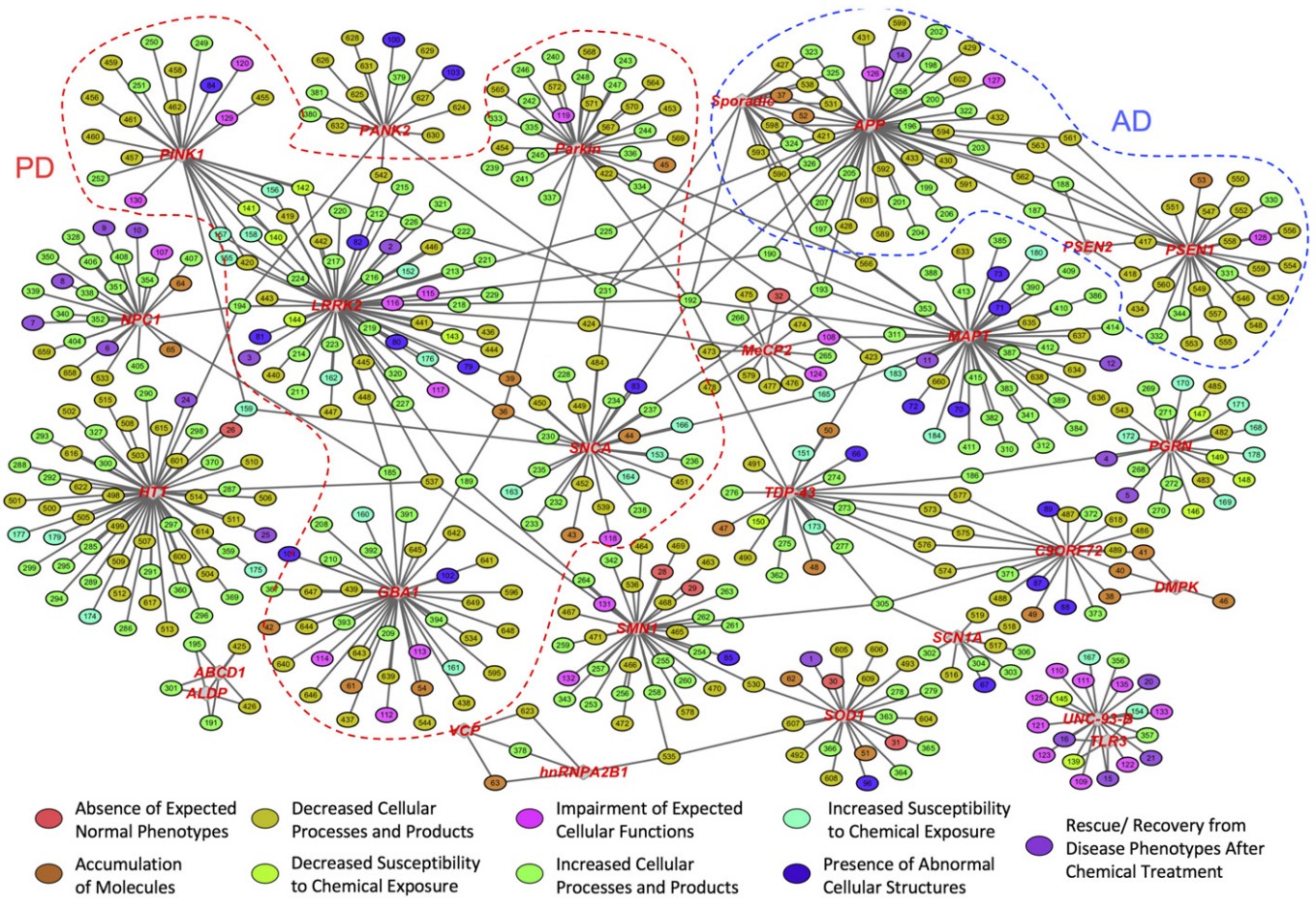

**Figure 5.  A network view of overlapping phenogenetic associations.**

The network view was built by combining genetic and phenotypic associations. Diamond nodes represent gene loci and are labeled in red while elliptical nodes indicate phenotypes, which are colored by their phenotypic class as described by Fig EV2. Each phenotypic node is represented by a distinct number and, for identification purposes, can be found in Appendix Table S10. The full version of this network, generated through an identical method of analysis, is included in Appendix Fig S3.

expression, such as to *CYP26A1* and *CXCR4,* which both are involved in the migration of progenitors in the developing brain (Stumm *et al*, 2003) (Appendix Fig S4G and H). Comparable to the observed molecular phenotypes, these cells show a great number of cellular phenotypes, which although is not causative, implies an association between alterations in molecular and cellular phenotypes.

We then determined the association between molecular and cellular phenotypes in neural cells to determine if genes linked to neurodevelopment and neurodegeneration would both show marked molecular abnormalities, reflecting the presence of their many cellular phenotypes. Indeed, we observed that NSCs with *LRRK2* mutations displayed abnormal molecular phenotypes, exhibiting upregulation of genes associated with apoptosis and nitric oxide processes (Appendix Figs S6 and S7). Finally, neurons from patients with *SNCA, SMN1,* and *DISC1* mutations show altered expression of genes involved with chromatin, survival, and genome stability (Fig 6A–E). For instance, neurons from schizophrenia patients with mutated *DISC1,* which is thought to have a neurodevelopmental dimension to its pathology (Walsh *et al*, 2008), show downregulation of metabolic pathways and *LAMA2,* a gene in

which *de novo* mutations have been identified in cases of sporadic schizophrenia (Xu *et al*, 2012) (Fig 6D and Appendix Fig S8). Unlike iPSCs from patients with neurodegenerative disorders, these mature cells exhibited an increased number of cellular phenotypes that correspond to the altered genes and pathways, further suggesting that these molecular changes may be the substrate of cellular phenotypes and maintain this developmental stage disparity (Appendix Fig S3 and Fig 4).

## Spatiotemporal localization of *in vitro* dysregulated gene expression in the human brain

Next, we asked if transcriptional dysregulation seen *in vitro* corresponded to a spatial and temporal expression patterning in the human brain, which could indicate that dysregulated gene networks observed within patient-derived cells *in vitro* correlate with a correct spatiotemporal localization of gene expression seen *in vivo* during the disease. Using the Allen Brain Atlas, we examined the expression pattern of the top 191 most dysregulated genes in diseased iPS-derived cells compared to controls from the available GEO datasets (Appendix Table S11). We performed a cluster analysis with the

**Table 1. Association of phenotype ontology and gene ontology.**

| Phenotype ontology term | Gene | P-value | Gene ontology functional annotation | P-value |
|---|---|---|---|---|
| SMN protein | SMN1 | $5.50 \times 10^{-15}$ | Absent | Absent |
| Aβ protein | PSEN1 | $6.66 \times 10^{-14}$ | Absent | Absent |
| Interferons | UNC-93-B | $9.33 \times 10^{-14}$ | Absent | Absent |
| TDP inclusions | TDP-43 | $9.63 \times 10^{-14}$ | Formation of cytoplasmic inclusions | $6.02 \times 10^{-4}$ |
| Neurofilaments | SOD1 | $4.79 \times 10^{-10}$ | Formation of neurofilament inclusions | $1.09 \times 10^{-4}$ |
| Caspase-4 | APP | $4.79 \times 10^{-10}$ | Absent | Absent |
| RNA foci | C9ORF72 | $1.12 \times 10^{-9}$ | Absent | Absent |
| Motor neurons | SOD1 | $3.31 \times 10^{-9}$ | Neurodegeneration of motor neurons | $3.28 \times 10^{-4}$ |
| Glutamatergic neurons | MeCP2 | $3.47 \times 10^{-9}$ | Absent | Absent |
| Binding immunoglobin protein | APP | $1.35 \times 10^{-7}$ | Absent | Absent |
| Cellular autophagy | NPC1 | $3.23 \times 10^{-7}$ | Autophagy | $1.66 \times 10^{-2}$ |
| Glutathione | PINK1 | $4.50 \times 10^{-7}$ | Absent | Absent |
| ATP levels | HTT | $5.17 \times 10^{-7}$ | Depletion of ATP | $2.74 \times 10^{-4}$ |
| Mitochondrial membrane | PINK1 | $5.69 \times 10^{-7}$ | Function of mitochondria | $7.66 \times 10^{-4}$ |
| Tau filaments | APP | $2.75 \times 10^{-6}$ | Generation of tau filament | $1.09 \times 10^{-4}$ |
| Nuclear morphology | LRRK2 | $1.16 \times 10^{-5}$ | Organization of nuclear envelope | $4.38 \times 10^{-4}$ |
| Neural rosettes | ATP7A | $7.51 \times 10^{-5}$ | Absent | Absent |
| GABAergic neurons | SCN1A | $7.51 \times 10^{-5}$ | Absent | Absent |
| Lamin | LRRK2 | $1.35 \times 10^{-4}$ | Absent | Absent |
| Aβ protein | APP | $1.85 \times 10^{-4}$ | Aggregation of amyloid fibrils | $5.47 \times 10^{-5}$ |
| Alpha-Synuclein | SNCA | $2.06 \times 10^{-4}$ | Absent | Absent |
| Increased susceptibility to chemical exposure | LRRK2 | $4.13 \times 10^{-4}$ | Absent | Absent |
| Caspase-3 activation | LRRK2 | $4.57 \times 10^{-4}$ | Absent | Absent |
| Cellular autophagy | GBA1 | $1.90 \times 10^{-3}$ | Absent | Absent |
| Motor neurons | SMN1 | $2.85 \times 10^{-3}$ | Loss of motor neurons | $2.74 \times 10^{-4}$ |

Table comparing phenotype ontologies and gene ontology functional annotations with respective P-values. If the phenotype ontology was not reported in the current gene ontology, it was termed "Absent".

hypothesis that the dysregulated gene networks of iPS-derived CNS cells *in vitro* belong to discrete areas of the brain associated with the pathology of the disease where endogenous CNS cells reside.

First, we focused on dysregulated genes from iPSCs. For example, the adult brain heatmap of iPSCs with *FXN* mutations shows increased expression in the myelencephalon and pons, two brain regions affected in patients (Koeppen & Mazurkiewicz, 2013) with Friedreich's ataxia (Appendix Fig S9C). Likewise, in the adult brain heatmap of iPSCs with mutated *SNCA*, a majority of the genes are highly expressed in regions associated with Parkinson's pathology, like the globus pallidus (Hardman & Halliday, 1999) (Appendix Fig S9F).

To determine if this localization was maintained in differentiated progeny, we next studied NSCs with mutant *LRRK2* and observed increased expression of dysregulated genes in the cerebellum (Seidel *et al*, 2017), mesencephalon, and myelencephalon (Qamhawi *et al*, 2015), where significant PD pathology is observed, particularly in the substantia nigra pars compacta, in the prenatal and adult brain heatmaps. Interestingly, this dysregulated gene expression localizes in the late months and years of the developmental transcriptome, consistent with the degenerative component of Parkinson's

pathology (Fig 7A–C). Last, we examined the localization of gene expression from patient-derived neurons with mutant *DISC1*. In line with the neurodevelopmental model of schizophrenia, the dysregulated genes are highly expressed in the cortex and SVZ of the prenatal brain and during the weeks of postconception, suggesting the involvement of this gene in neural progenitors and the microarchitecture of the brain. Similarly, in the adult human brain, there is high gene expression in the CA1 region of the hippocampus, where disease pathology is found (Schobel *et al*, 2009) (Fig 8A–C). Altogether, these data suggest that the neuropathological behavior of these dysregulated networks is faithfully encoded in patient-derived cells *in vitro*, spanning from iPSCs to neurons.

**A web database for the phenogenetic map from iPSC-derived cellular phenotypes**

To provide this information to the scientific community, we designed a web platform: www.iPhemap.org that will make the curated collection of phenotypes available to the public (Web Resources EV1). This web platform has a user-friendly interface that can be searched for all reported cellular phenotypes, in addition to

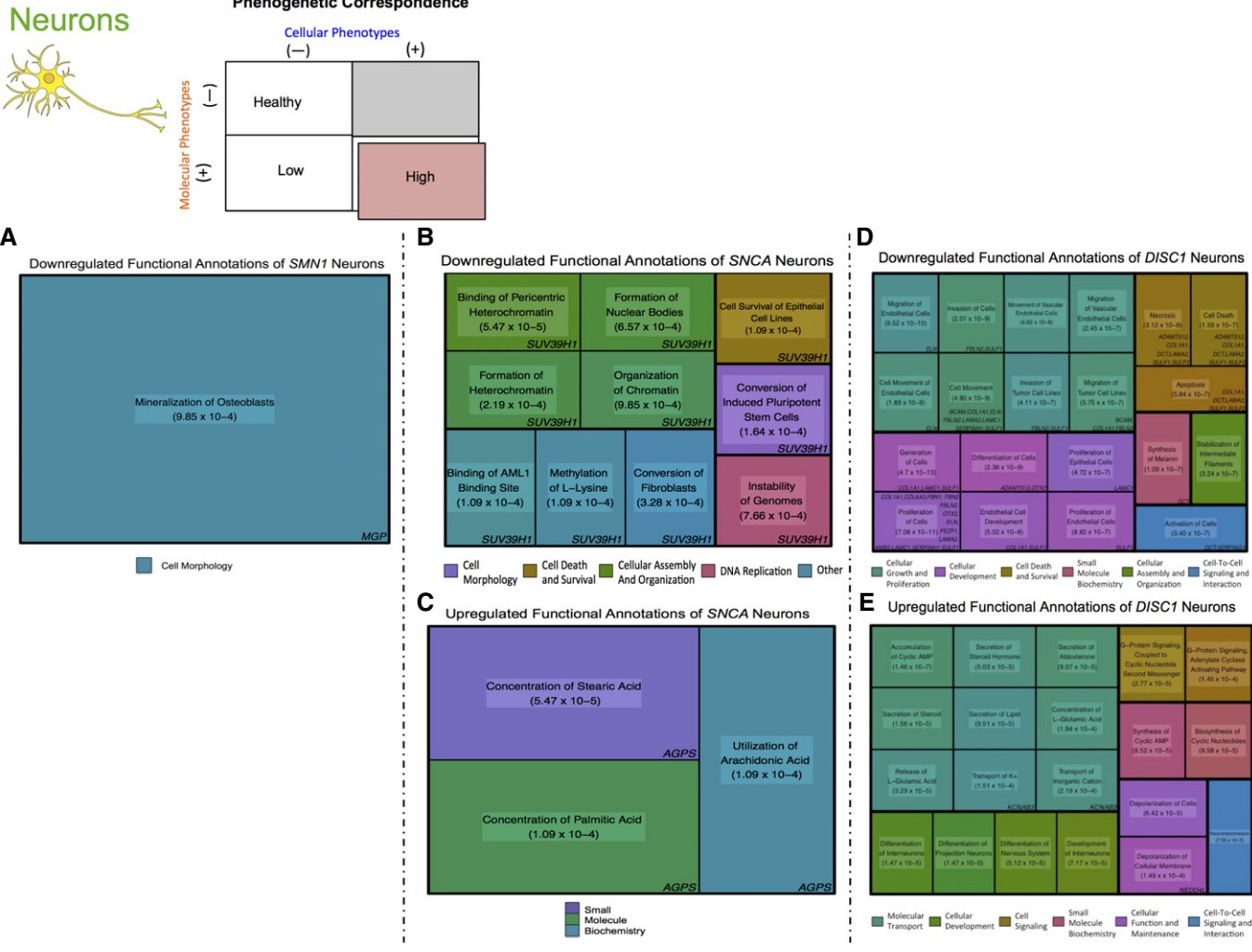

**Figure 6. Phenogenetic correlation of neurons in neurodegenerative and neurodevelopmental disorders.**

Representative treemaps of genes linked to neurodegenerative and neurodevelopmental diseases show significant molecular alterations at the neuronal level, reflecting the presence of reported cellular phenotypes and therefore high phenogenetic correlation.

A       Downregulated functional annotation of neurons with mutant *SMN1*.

B, C    (B) Downregulated and (C) upregulated functional annotations of neurons with mutated *SNCA* show decreased expression of *SUV39H1*, a regulator of neuronal survival (Liu *et al*, 2005), and upregulation of *AGPS*, a gene involved in lipid biosynthesis (Brites *et al*, 2004), respectively.

D, E    Neurons containing *DISC1* mutations show (D) downregulation of genes associated with neurogenesis, like *OTX2* (Puelles *et al*, 2004), and (E) upregulation of gene expression, including *NEDD4L*, related to neurotransmission (Laedermann *et al*, 2013).

molecular phenotypes when available (Web Resources EV1). iPhemap will be continually updated by the database's curators with the goal of maintaining all the phenotypes reported from iPSC models of neurological diseases.

## Discussion

The field of iPS modeling of neurological diseases is a nascent field that continues to evolve with improved and novel methodologies, which introduces variation to practices and techniques among the field. One might argue that a field synopsis is premature given the early state of the iPSC field; however, we posit that due to lack of homogeneity, a systematic assessment of the practices and

accumulated knowledge of phenotypes in this field are necessary. Here, we report a systematic analysis of the correlation of 663 neuronal phenotypes with genotypic data from 243 patients and 214 controls, creating a public repository that catalogs current CNS cellular phenotypes in the field of patient-derived models of neurodegenerative and neurodevelopmental disease (http://www.iPhemap.org). Our analysis provides a taxonomy of phenotypes from patient-derived iPSC models of neurological diseases and their distribution by developmental stage. The network we generated not only shows overlapping phenotypes and the degree of association between cellular and molecular phenotypes, but also illustrates opportunities to develop better phenogenetics of iPSC-derived cells associated with human pathology. Our web resource provides a tool for cataloging the phenogenetic correlations of human neurological

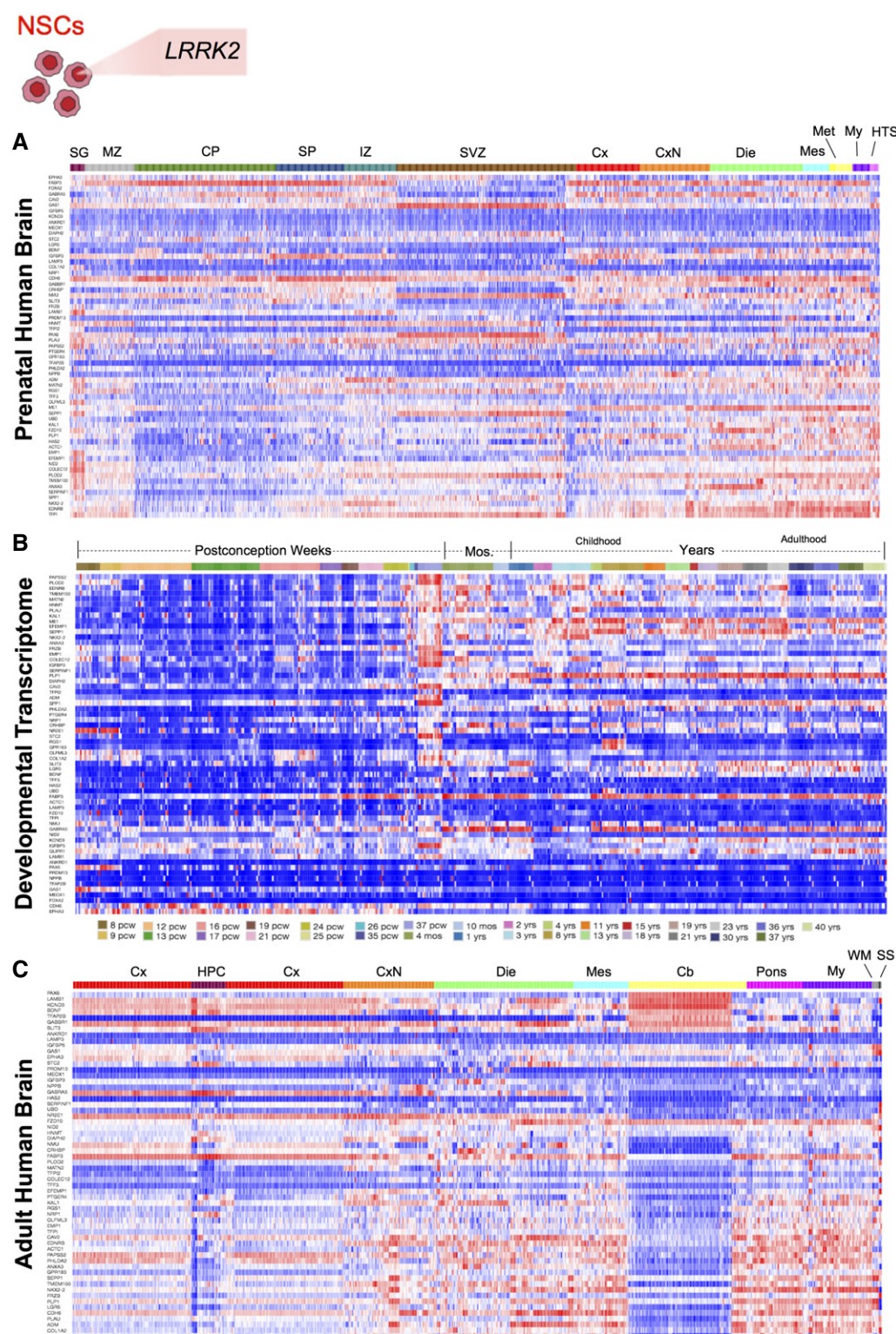

**Figure 7.  Spatial and temporal expression patterning of dysregulated genes from NSCs *in vivo*.**

A–C  Heatmaps show localization of dysregulated gene expression from NSCs with mutant *LRRK2* in the (A) developing cortex (CP) and progenitor zones (IZ, SVZ) of the prenatal human brain, (B) temporal expression predominates in the brain during the late months of development through the years of adulthood, and (C) spatial gene expression in the adult human brain localizes to the mesencephalon (Mes), myencephalon (My), and cerebellum (Cb). CP, Cortical plate; Cx, Cortex; CxN, Subcortical Nuclei; Die, Diencephalon; HPC, Hippocampal Formation; HTS, Hindbrain transient structures; IZ, Intermediate zone; Met, Metencephalon; Mos., Months; MZ, Marginal zone; SG, Subpial granular zone; SP, Subplate zone; SS, Sulci and spaces; SVZ, Subventricular zone; WM, White Matter.

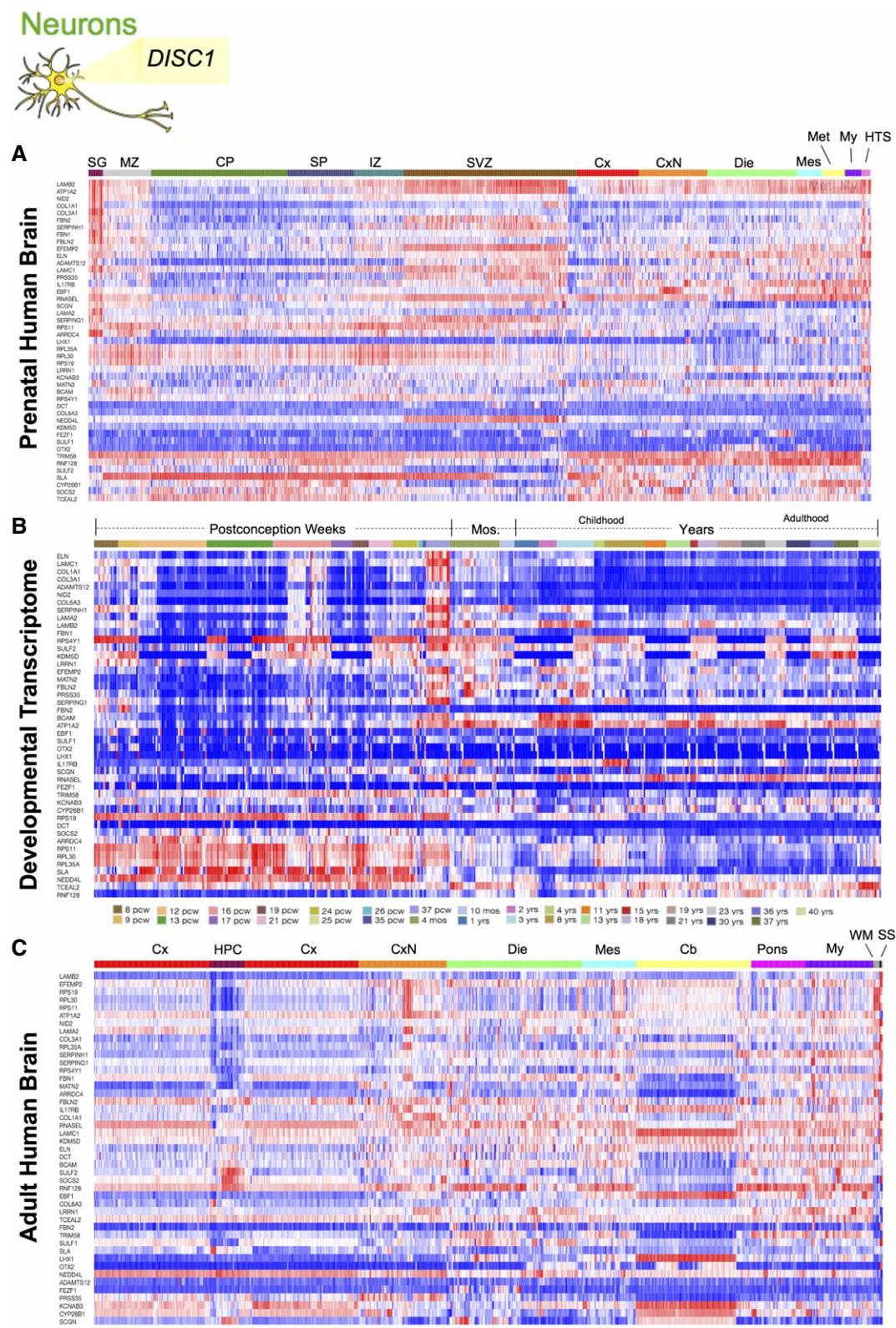

**Figure 8.   Spatial and temporal expression patterning of dysregulated genes from neurons *in vivo*.**

A–C    Heatmaps show localization of dysregulated gene expression from neurons with mutations in *DISC1* in the (A) developing cortex (CP, SP) and progenitor zones (SVZ) of the prenatal human brain, (B) temporal expression predominates in the early and late weeks of postconception brain during development, and (C) spatial gene expression in the adult human brain localizes to the subcortical nuclei (CxN). CP, Cortical plate; HTS, Hindbrain transient structures; IZ, Intermediate zone; Mos, Months; MZ, Marginal zone; SG, Subpial granular zone; SP, Subplate zone; SS, Sulci and spaces; SVZ, Subventricular zone.

diseases modeled by iPSCs and a platform to track the relevant, novel CNS cellular phenotypes to improve upon existing phenotypic assays for drug development and to better understand human disease pathogenesis.

The present analysis includes work that fulfilled stringent inclusion criteria and highlights the need for a set of criteria of reproducible, minimal information for reporting iPS cell models, given the experimental noise imposed by lack of standardized protocols in the field (Brennand *et al*, 2015).

**The need for minimal information about iPSC experiments (MiPSCE) in neurology**

One hurdle for modeling neurological diseases with iPSCs is the use of the molecular and cellular phenotypes obtained from these cells as reproducible and scalable metrics to discover pathways *in the dish* for therapeutic purpose. Our analysis has proven that there are phenotypic differences between patient-derived CNS cell types compared to controls, which can be reproducible, especially the most frequently assessed. From the 93 primary studies generating iPSCs included in our analysis, 35% were conducted with three or more patient cell lines, 42% with two lines, and 23% were done with only one line. Likewise, the number of cell lines derived from each patient varied markedly as well, all supporting the need for minimal standards in the reporting of iPSCs models.

To further develop the phenogenetics of iPSC-derived cell types, experiments must test the sensitivity and specificity of cellular phenotypes for a particular cell and disease to assess the minimal number of patients and cell lines needed to reach definitive conclusions. These efforts cannot be realistically undertaken by a single laboratory and may need the efforts of consortia and scientific associations in stem cell research. We do anticipate though, that as several iPS cell banks continue to grow, laboratories will have increased access to cell lines, enabling the majority of future experiments to make use of more than one diseased and control line to increase the robustness of their findings. In our analysis, we observed a lack of uniformity in not only the methodology utilized, such as differences in cell culture conditions, but also in the reporting of iPSC differentiation experiments, including the documentation of fate, yield, and purity of the derived cell types. The future inclusion of such data from differentiated cultures may help address the need for a standard set of criteria to define a given cell type, perhaps with thresholds of purity defined by marked expression and physiological measures, which would increase the reliability of comparing reported phenotypes, for it is unclear whether these differences can affect the phenotypes and gene expression of the derived cells (Fig 1). As such, the establishment of standards will improve reproducibility and standardize methodologies among different laboratories.

We extracted seven categories that comprise the minimal information we found useful throughout curation and suggest their adaptation for future iPSC studies (Fig 1). Based on our analysis, we have established this minimal information that should be included in all future studies, which also integrates prior efforts to homogenize iPSC field practices (Luong *et al*, 2011) (Materials and Methods and Appendix Table S12). In addition to our suggested MiPSCE, we have proposed that future work in the iPSC field will leverage big-data techniques in a community wide effort to establish reliable and comparable datasets, allowing for researchers to draw conclusions of the phenogenetic nature from multiple iPSC lines (Del Sol *et al*, 2017).

In general, improved measures of phenotypic assays and standardizing culture conditions in iPSC experiments would enhance phenotype analyses of human cells, a strategy that has been successful in the phenogenetics of *Caenorhabditis elegans* and *Arabidopsis thaliana* (Kuromori *et al*, 2006; Atwell *et al*, 2010). In the future, by accumulating more phenotypes in mutated cells from human neurological diseases, we can build more complete phenogenetic maps. This is crucial as our current early phenogenetic map contains an unavoidable, inherent bias toward diseases and mutations that were more frequently investigated in the literature. Furthermore, there may be a bias in which phenotypes were probed for due to assays that were better adapted for use in iPSCs or phenotypes that have previously been reported in animal, postmortem, and primary cell culture studies (Fig 1A). This potential bias may have influenced our current set of overlapping phenotypes as investigators may have been more inclined to test for phenotypes based on prior work, thereby diminishing the potential for phenotypes to link genes from different diseases. Therefore, expanding which phenotypes are tested for, outside the scope of past studies, will further enrich phenogenetic analyses.

**Utility and limitation of the atlas and translational challenges of the iPSC phenotype field**

Our phenogenetic map is limited to neurological diseases caused by somatic mutations; thus, it should be considered an early effort that will be enriched and refined by additional work of the field as evidenced by the evolution of other mapping efforts (Kuromori *et al*, 2006; Atwell *et al*, 2010), such as the inclusion of complex genetic disorders caused by copy number variants (CNVs), single nucleotide polymorphisms (SNPs), and other low-penetrance mutations. It will be very difficult to anticipate if *in vitro* patient-derived models will ever replace other models of neurological diseases; however, human iPS modeling could have practical translational utility. For instance, the identification of overlapping phenotypes among diseases that are thought to have distinct pathologies would reveal mutually, disrupted cellular processes that may be responsive to similar therapies, indicating that a single phenotype can be used as a potential biomarker, an "inter-disease biomarker," for diverse *in vitro* models of neurological diseases. Furthermore, the elucidation of concordant phenotypes within a specific disease would allow for the anticipation of disease phenotypes in a patient when the specific genetic mutation is unknown. These concordant phenotypes may be used to create phenotypic assays to detect a particular disease *in vitro*, as an "intra-disease biomarker". For example, the *increase in Aβ* observed in all of the mutations linked to Alzheimer's disease can also be seen in late onset Alzheimer's disease (LOAD) and while this finding may not be surprising, it does, however, demonstrate that concordant iPSC phenotypes are robust tools for studying disease. Comparably, an *increase in α-synuclein*, reported in the majority of the mutations linked to Parkinson's disease, could be utilized as a predictive phenotype signature in the sporadic disease-induced cells.

One critical question in iPSC modeling is if it is more relevant to replicate findings to increase the sensitivity or specificity of *in vitro* phenotypes as biomarkers or to find novel phenotypes with unknown specificity. Although costly, confirming the presence of disease-specific phenotypes in multiple cell lines with distinct genomes will reduce the contributions of experimental noise and limit the effect of spurious variation expressed by a single line. It is impractical for a single group to undertake such a prospective analysis, rather a concerted effort through a consortium may have the necessary resources. Additionally, the continued practice of our retrospective assessment of phenogenetic "level of evidence," defined as the number of cell lines with different mutations in a gene expressing a particular phenotype, may also help in validating disease-specific phenotypes (Appendix Table S10). Moreover, investigation of the relationships between sporadic and existing mutation-induced phenotypes can help to reveal important mechanistic

information about sporadic diseases, especially when the intrinsic neuronal mutation has yet to be established, like in multiple sclerosis (Douvaras *et al*, 2014; Orack *et al*, 2015), or in diseases where CNVs provide a modest risk factor for susceptibility, like autism, or mental disorders, like schizophrenia (Brennand *et al*, 2011; Wen *et al*, 2014; Srikanth *et al*, 2015).

Our analysis and taxonomy serve as a potential resource for tracking the most relevant cellular and molecular phenotypes in modeling neurological diseases using iPSCs and could inform future strategies to regulate pathways altered at the cellular level *in vivo* through pharmacological targeting of disease-associated traits. More importantly, a catalog of the ever-increasing number of mutant phenotypes into a new taxonomy of iPSC-derived phenotypes will aid future large-scale phenotype analysis in neurological disorders by correlating multilayer -omics information from the clinical, radiological, cellular, and molecular data of patients. Our analysis will

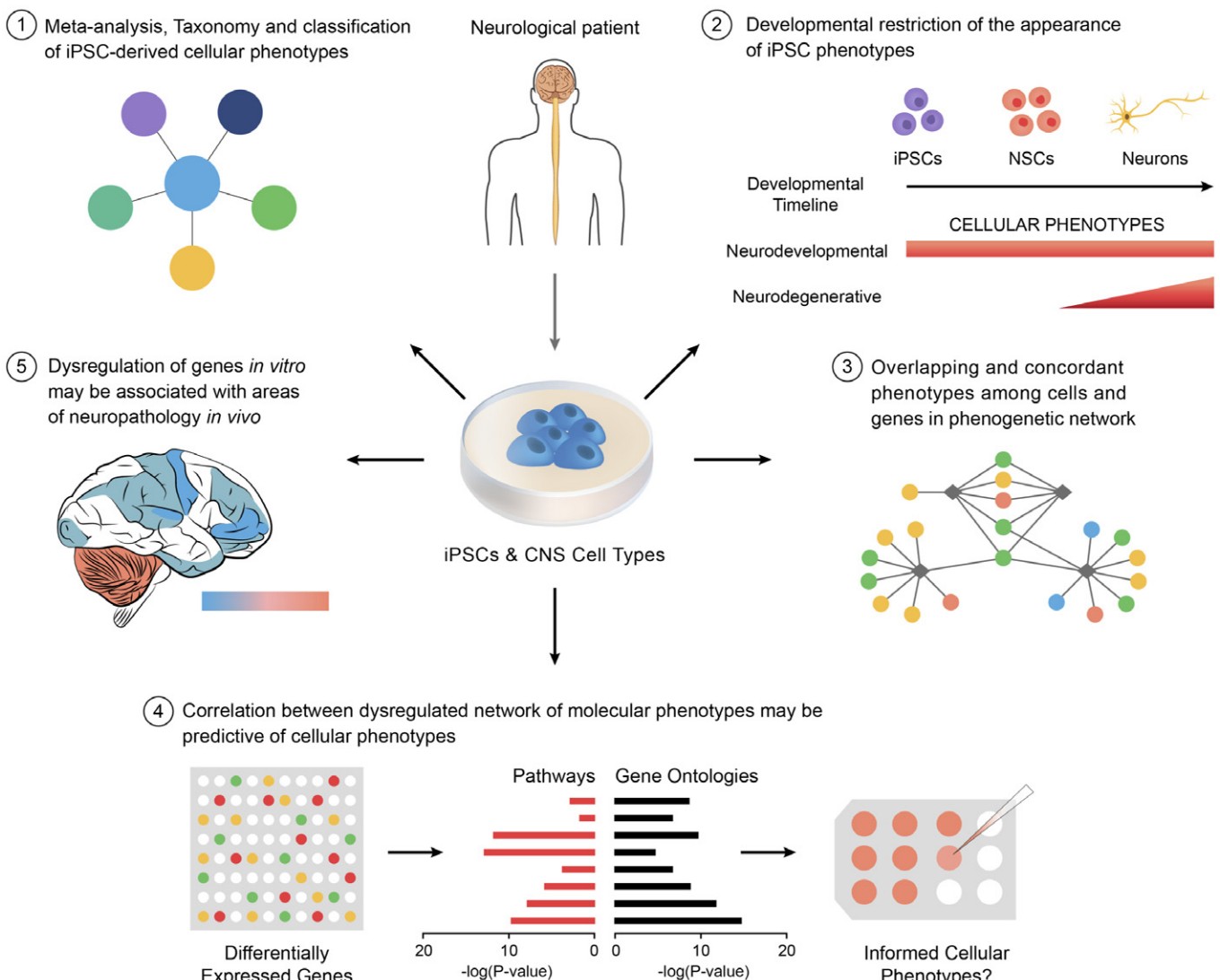

**Figure 9.   Novel principles of phenogenetic correlations of iPSC-derived cellular phenotypes derived from patients included in our meta-analysis of iPSC models of neurological disorders**.

also be useful as we build novel algorithms to determine the robustness of iPSC results, including factors such as power, validation, and replication by independent assays, which over time can reject spurious phenotypes from the less optimally designed experiments. The future of using patient-derived neural cells should aim to develop and study the iPSC phenotypes that are relevant for human neurological diseases. Therefore, the further refinement and replication of induced neural disease phenotypes are critical for modeling the spectrum of neurological diseases.

### Conclusions

In summary, we examined the current iPSC field practices of modeling neurological diseases and carried out a comprehensive analysis of the phenogenetic relationships of patient-derived cellular models of neurological diseases and catalogued them in the iPhemap database. Although the results obtained in our analysis are retrospective, our findings illustrate the presence of diverse practices within the field. We posit that a retrospective analysis, like a mid-term examination, now 10 years after the discovery of iPSCs, is needed to improve the reproducibility of the field, especially as we are now investigating the role of genetic variation, such as SNPs, in genomewide association studies (GWAS) (Sweet, 2017) of *in vitro* iPSC phenotypes using multiple patients. Importantly, we identify areas of opportunity to improve the reproducibility of experimental results and to increase the translational utility of models. Finally, we propose, for the first time, a set of principles for the phenogenetic analysis of *in vitro* models of human diseases with iPSCs (Fig 9), which may be expanded and revised with future work (Movie EV1).

## Materials and Methods

### Methods summary

We examined a total of 93 iPSC studies in modeling neurological diseases including neurodegenerative and neurodevelopmental disorders that fulfilled the inclusion and exclusion criteria. The inclusion criteria were as follows: (i) Studies that used iPSCs derived from human patients to investigate cellular phenotypes caused by neurological diseases. (ii) Studies that specify disease and gene mutation of all iPSCs and any additional differentiated cells under investigation. (iii) Studies that specify type of control cells lines utilized. (iv) Studies that describe phenotypic differences in comparison with their respective control cells. The exclusion criteria were as follows: (i) Studies that used non-iPSC-derived cells as controls, such as embryonic stem cells. (ii) Studies that introduced disease mutations into otherwise healthy iPSCs. (iii) Studies that used iPSC-derived cells with low-penetrance mutations, like CNVs or SNPs. (iv) Studies that only reported gene expression profiles, including microRNAs. (v) Studies that modeled iPSC-derived organoids. We first analyzed the experimental methodologies of these studies and then documented all of the phenotypes from diseased iPSC-derived cells, which we organized into nine distinct phenotypic categories. From this, we generated a Circos plot and ideogram to illustrate the phenotype-category relationships. Moreover, to further explore the associations between our 663

phenotypes and examined genes, we generated a phenogenetic map and conducted statistical analyses, to discover new phenogenetic associations. In addition to the cellular phenotypes, we also conducted analysis of microarray data on eleven of the studies to elucidate molecular phenotypes, associated with transcriptional dysregulation. Following our meta-analysis, we developed an online web tool, titled iPhemap, which is a curated repository of iPSC disease phenotypes and allows users access and to submit potential data to our phenogenetic database.

### Search strategy and meta-analysis

We began our search through the published iPSC articles by utilizing PubMed and specific pertinent keywords. Our initial search employed the following keywords: neurodegenerative, disease, human iPSCs, and iPSCs. This resulted in a large return of articles, $n = 36$. Next, we conducted a narrower search, which consequently expanded our candidate article number to 52. This second search introduced common neurodegenerative diseases and their gene mutations (i.e., Parkinson's and *LRRK2*) with the words iPSC and human iPSC to better address the desired article content by our search terms.

Through our close reading of these reports, we discovered 25 additional articles for consideration. We then established a list of quality control criteria for our meta-analysis, including: (i) Articles that used iPSCs derived from human patients to investigate cellular phenotypes caused by neurological diseases. (ii) The disease and gene mutation of all iPSCs and any additional differentiated cells under investigation are clearly stated in the article. (iii) Diseased iPSCs and/or differentiated cells under investigation are compared against control cells lines of the appropriate cell type. (iv) Diseased cells under investigation exhibited phenotypic differences in comparison with their respective control cells. In addition to the inclusion criteria, we formulated specific exclusion criteria: (i) Articles that used non-iPSC-derived cells as controls, such as embryonic stem cells. (ii) Articles that introduced disease mutations into otherwise healthy iPSCs. (iii) Articles that used iPSC-derived cells with low-penetrance mutations, like CNVs or SNPs. (iv) Articles that only reported gene expression profiles, including microRNAs. (v) Articles that modeled iPSC-derived organoids. To expand our analysis across the entire field of neurological disease and update our bevy of papers with those recently published, we performed a final search, which returned 36 additional articles, thus increasing our total to 113 papers.

However, upon further examination, we omitted twenty candidates from our original compilation of 113 as they failed to meet our requirements. We then analyzed these 93 articles to document disease-specific phenotypes in iPSC-derived cells that were different in comparison with their control lines. Our analysis included studies with 32.2% of manuscripts being published in journals of impact factor (IF) of 5–10, 17.2% in 10–20 IF, and 26.9% of more than 20 IF. To document the pertinent information from our meta-analysis, we recorded all observed phenotypes, with the same lexicon annotated in the papers, gene alterations, and corresponding disease names from the accepted papers. Moreover, to organize the large number of phenotypes, we established nine separate phenotypic categories. These categories served to highlight phenotypic patterns throughout diseases. Upon completion of this initial analysis and

organization, we curated the 93 accepted articles three different times by three separate investigators to ensure accurate phenotypic extraction had been conducted.

## Minimal information categories

Despite our inclusion and exclusion criteria, the examined 93 articles contained heterogeneity within their provided methods. Even with comprehensive descriptions and extensive information provided, discrepancies across our group of articles remained. Within the articles, seven categories consistently arose and make up the minimal information which we propose should be provided in all future studies (Appendix Table S12). The seven categories and their prominence are as follows:

*Clinical information of patients and primary fibroblast isolation (63.4%)*
The isolation of primary patient fibroblasts from diseased patients is necessary for all studies in the iPSC field. In accordance with the papers, we analyzed general clinical information of patients from which the fibroblasts (or other cell type that needs to be specified) are taken should be provided, including information such as age, gender, disease under investigation, and age when disease appeared, if known. If fibroblasts, or iPSCs, are received from a third-party cell repository, the name of the distributor from which the cells were received as well as the line and or patient number should be provided in the article as well for reference.

*Generation of iPSCs (87.1%)*
With the existence of several different ways to drive cells toward pluripotency, the procedure by which each iPS cell was induced should be provided. This includes any manipulation of the primary cell line to reach the expression of pluripotent genes (i.e., retroviral infection procedures and confirmation of gene expression), as well as quality control methodologies for ensuring pluripotency has been achieved, like teratoma formation.

*Detailed cell culture/maintenance information (86.0%)*
The environment and medium on which the iPSCs and any differentiated cells were cultured and maintained on should be provided in order to allow researchers access to all information pertaining to the growth and upkeep of the cells being studied.

*Detailed differentiation of iPSCs to any cell type (88.2%)*
The process by which any cell type is derived from the existing iPSCs should be stated clearly in each article to provide accurate details as to how it was achieved. This should include the medium the cells are placed on as well as the different culture conditions to obtain mature cells, genes, or markers used by the researchers to properly confirm the identity of the cells after they underwent differentiation. The purity of the iPSC-derived progeny cultures, defined by differentiation markers and other physiologic measures, should also be included.

*Validation of the mutation being studied (52.7%)*
The gene mutation being studied by the researchers should be already stated in the paper with the disease it pertains to in the isolation of the fibroblast, but researchers should also provide the

means by which they confirmed the retention of the cells' mutation after iPSC generation and/or differentiation into other cell types used for their study.

*Gene delivery methods for cells if used (100% of papers utilizing gene editing)*
Papers should continue to provide all information regarding the alteration in genes, including already mutated genes, to create a secondary control line or any line that may be used as a comparison with the diseased lines. This includes processes such as lentiviral infections, episomal plasmids, TALEN mediation recombination, and zinc finger nucleases.

*Procedures of assays/specific phenotypic search methods (95.7%)*
These procedures suggested would contain any type of test performed on the iPSCs and differentiated cells in the study to determine a phenotype to be specific to the disease or not specific to the disease. This includes procedures used to measure protein levels as well as procedures used to view disrupted cellular structures, these should include numbers of technical and biological replicates.

## Assumption of the phenogenetic model

After the curation and extraction of all relevant phenotypes, we formulated a phenogenetic model. Our model posits that by using highly curated phenotypic information from patient-derived cells with somatic mutations, we can build a phenogenetic correlation for each phenotype and genotype using the following relationship: Phenotype ($p_i$) is a function of genotype ($g_i$) plus an environmental component ($e_i$).

$$p_i = f(g_i) + e_i$$

$i$: individual patient-derived cell; $g_i$: genotype of $i$; $p_i$: Quantitative phenotype of $i$ cell: Cellular phenotypic trait (CPT); $e_i$: Environmental contribution to $p_i$.

The environmental component was excluded for two reasons: (i) It is not possible with the data obtained to measure the influence of the culture environment. In addition, since the analysis is not on individual cells, but a group of cells, this could equalize the potential effects of the *in vitro* cell culture environment, although we are aware that the culture environment could influence cellular phenotypes, given the myriad of protocols for iPS cell generation reported. (ii) The assumption of our model, based on the results from patient-derived cells with pathogenic mutations, is that the cellular phenotypes obtained from these cells may represent highly disruptive alterations in a cellular network, as supported by the hundreds of abnormal phenotypes observed that suggest some mutant phenotypes may supersede any variation induced by culture environment. Rather, these phenotypes are caused by highly pathogenic and penetrant mutations with a high degree of causality, more than what may be seen in iPSC models derived from cells with smaller, discrete CNV or SNP, which we excluded from our analysis. Therefore, the equation was simplified to:

$$p_i = f(g_i)$$

$i$: individual patient-derived cell; $g_i$: genotype of $i$; $p_i$: Quantitative phenotype of $i$ cell: Cellular phenotypic trait (CPT).

### Circos plot generation

To showcase the phenotype-category and phenotype-cell type relationship, we organized the related data in a Circos plot. We initially gathered the proportional data for the number of phenotypes within each of our nine categories along with the number of observed phenotypes for each cellular type. From these proportions, we calculated numerical values, which were organized into a table (Appendix Table S3). We then uploaded this table into the Circos plot generator at circos.ca. http://mkweb.bcgsc.ca/tableviewer/visualize/ (Krzywinski *et al*, 2009). We predominantly employed the default settings for this program except for several small changes: small ribbons on top of large, color by column, order by column first, normalization of ribbon sizes, and light gray transparency (3) of Q1, Q2, and Q3. The software denoted the percentages and proportions of each cell type to the respective phenotypic category, which resulted in a more robust display of the data collected.

### Ideogram generation

In order to display the location of each gene paired with its phenotypic categories, we sought to generate an ideogram. By identifying the phenotypic categories observed for each gene, we utilized the PhenoGram program: http://visualization.ritchielab.psu.edu/phenograms/plot to produce the desired ideogram (Wolfe *et al*, 2013). We used the default settings of this program, which included the use of the human genome and human cytobands.

### Developmental stage and phenotype analysis

Using the phenotypes and genes recorded for each cell type in the heatmap (Appendix Fig S1), the percent of reported phenotypes for each patient-derived CNS was manually extracted. GraphPad Prism Software was then utilized to generate the plots depicting phenotype distributions of diseases by cell type, phenotype by gene, and gene by phenotype (Fig 4). One-way analysis of variance (ANOVA) with Bonferonni multiple comparisons tests (Fig 4B–F) and two-way ANOVA with Tukey's multiple comparisons test (Fig 4G) were performed.

### Network generation

From the data provided by our meta-analysis, we elucidated the overlapping phenotypes, which we recorded in addition to phenotypes only expressed by a single locus. To generate a network, we formatted our relationship data into a table of source nodes (loci) and their target nodes (observed phenotypes). We then uploaded our processed data table to generate a network of nodes and edges through the Cytoscape application. By doing this, we generated edges when a locus expressed a phenotype, which indirectly connected genes through an overlapping phenotype (Shannon *et al*, 2003). We then colored each phenotype node based upon its corresponding phenotypic category.

We dictated the layout of our network by employing a force-directed paradigm, which utilizes an algorithm to position nodes based on a physics simulation of spring-like forces to generate an aesthetically pleasing layout. From our overarching network, we generated more nuanced networks to showcase only the overlapping phenogenetic network and specific disease-phenogenetic networks for genes associated with PD and AD.

### Statistical analysis of network

To illustrate that our generated network followed a power-law distribution, the node degree distribution graph was fitted with a power-law curve by the Network Analyzer application of Cytoscape and returned the equation $y = 53.358x^{-1.160}$ for the fitted curve, including the following statistical parameters: $R^2 = 0.717$ and $r = 0.922$. Furthermore, to statistically illustrate that all of the scatter plots followed a power-law distribution; we converted the axes of the all of the scatter plots to a logarithmic scale. Then, we performed linear regression analyses to calculate the $P$-value of each respective plot. As mentioned above, through our conversion of these axes into a logarithmic scale, the significant $P$-values demonstrated that our data follow a power-law distribution, which is typical of biological networks. To conduct all of the aforementioned statistical tests, we utilized the R statistical computing software (Assenov *et al*, 2008; RC Team, 2010).

### Ontology methods

We conducted functional annotation analyses with respect to the phenotypes involved in iPhemap. We calculated the phenotypic enrichment for each gene utilizing a Fisher's exact test, which compared the number of phenotypic observations with those directly observed by a particular gene, thus determining whether or not a particular phenotypic annotation was more significant for a gene. To account for the number of individual hypothesis tests conducted, we performed a Benjamini–Hochberg multiple comparisons test for all $P$-values to control for false discovery rate. We then termed the significant relationships established from our phenotype to gene approach as phenotype ontology.

We further analyzed the genes involved in these significant phenotypic relationships through the more common approach of gene ontology. We entered these genes into a gene ontology database to observe which functional annotations were statistically significant. Furthermore, we compared the related gene ontology functional annotation $P$-values with the $P$-values generated through our aforementioned phenotype ontology. To denote phenotypic ontologies that failed to share a corresponding functional annotation from the genome ontology, we termed them as "Absent". Thus, these phenotypic ontologies can be considered novel for each gene.

### Treemap and pathway generation

We also examined each of the accepted 93 articles in an attempt to determine if microarray analysis or another type of transcriptome profiling had been conducted, made publicly available as a GEO dataset. If so, which was the case for 24 studies, we next determined whether or not the microarray was conducted between the control and mutated iPSC-derived cells. This resulted in 22 candidate articles, but to ensure reproducibility of results, we inspected their protocols to determine if at least three samples of each control and

patient line had been utilized, which resulted in 18 papers that fulfilled this requirement. Furthermore, we required that the available GEO datasets were compatible with the GEO2R web-based interface or the *GEOquery* R package, thereby reducing our candidate pool to 13 papers. Last, we returned to the original text of each manuscript to ensure that a similar analysis had not been previously performed, which further diminished our pool to 10 studies. From these 10 studies, we processed the available molecular profile data through the GEO to identify dysregulated genes.

However, for some studies the transcriptome analysis resulted in a large number of unrelated differentially expressed genes. Therefore, we established specific parameters for these results ($P < 0.05$ and FC > $\pm2$) to narrow the scope of this analysis, which diminished the pool of studies to nine. To determine the associated molecular phenotypes and pathways, we entered the dysregulated genes into IPA. We then instituted further parameters to highlight the most pertinent molecular phenotypes and pathways, which include considering the functional annotations and pathways from the most significantly dysregulated gene network and establishing a statistical parameter of $P < 0.001$. Additionally, we exported the tables of functional annotation and pathway data provided by IPA to construct the assortment of pathway figures and treemaps by using the R treemap package (Tennekes, 2014) with the twenty most significant molecular phenotypes according to *P*-value.

### Heatmap generation

We also utilized the differentially expressed genes gleaned from our prior analysis of studies with publicly available microarray data that met our aforementioned criteria (see Treemap and pathway generation) to determine if the temporal and spatial expression of dysregulated genes correlated with typical disease pathology. To accomplish this, we made use of the Allen Brain Atlas, specifically the Allen Human Brain Atlas (Hawrylycz *et al*, 2012) and the BrainSpan Atlas of the Developing Human Brain (Miller *et al*, 2014). With the differentially expressed genes as input, we searched each respective database with default parameters to generate heatmap data for the temporal expression of genes in the developmental transcriptome, and spatial expression in the prenatal and adult human brain. Finally, we made use of Morpheus, a web-based matrix visualization software, to perform clustering analyses and generate heatmap images (https://software.broadinstitute.org/morpheus/).

**Expanded View** for this article is available online.

## Acknowledgements

The authors thank Drs. Christopher A. Walsh and David M. Panchision for helpful suggestions and Drs. Sally Temple, Thomas Kiehl, Huda Zoghbi, Michael K. Racke, Sarah Starossom, and Santosh Kesari for their critical reading of the manuscript. The authors also thank all the anonymous reviewers that evaluated the paper for their important insights and Sachin Rudraraju, Troy Hoffman, Nicolas Roe, and Sriram Durvasula for their assistance and technical expertise in the preparation of the web tool.

## Author contributions

EWH, JEV, JCO, CS, and FW collected and mined primary data. EWH, JEV, JCO, CS, JK, FW, and JI curated all data. EWH, JEV, JCO, CS, JK, SBL, MEH, FW, KSK, and JI interpreted all results. EWH and JI performed the computational analysis of data, conceptualized the idea, and designed the web tool. EWH, JEV, SBL, MEH, FW, KSK, and JI contributed to the writing of the paper.

## Conflict of interest

The authors declare that they have no conflict of interest.

## The paper explained

### Problem

Human disease modeling with iPSCs has enabled researchers to study the disease phenotypes of patient-derived cells directly in the laboratory. Now a decade after the discovery of iPSCs, hundreds of patient cell lines and neurological disease phenotypes have been generated, yet this abundance of information has become increasingly difficult to make sense of. Moreover, it is unknown how the research practices for iPSC neurological disease modeling vary among different laboratories. Synthesizing all this information to understand the phenotypic role of disease-promoting genes and identifying the limitations of our current practices will be crucial steps toward achieving the great translational potential of iPSC models of neurological diseases.

### Results

Our results indicate the diverse practices of the iPSC neurological disease modeling field. From our retrospective analysis of the published literature, we have developed a taxonomy of CNS cellular phenotypes and revealed that there are previously unrelated genes that show similar disease phenotypes. This work also showed that alterations in patient-derived cells at the level of gene expression correlate with the reported cellular phenotypes and these dysregulated genes are highly expressed in specific regions of disease in the human brain.

### Impact

This study identifies areas of opportunity to improve the reproducibility and translational utility of these neurological disease models and offers novel insights into the phenotype–genotype relationships of human disease models using iPSCs. As a valuable resource for the research community, this study provides a public, online database for researchers to query and deposit the most current phenotypic information from iPSC models of neurological diseases.

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
