## [Review Process File · EMBO Molecular Medicine]

iPhemap: An atlas of Phenotype to genotype relationships of human iPSC models of neurological diseases

Ethan W. Hollingsworth, Jacob E. Vaughn, Josh C. Orack, Chelsea Skinner, Jamil Khouri, Sofia B. Lizarraga, Mark E. Hester, Fumihiko Watanabe, Kenneth S. Kosik and Jaime Imitola

Corresponding author: Jaime Imitola, The Ohio State University

Review timeline:

Submission date:	21 June 2017
Editorial Decision:	22 August 2017
Revision received:	08 September 2017
Accepted:	12 September 2017

Transaction Report:

Editor: Céline Carret

1st Editorial Decision

22 August 2017

Thank you for the submission of your manuscript to EMBO Molecular Medicine. At long last, I'm happy to say that we have received the three reports from the two referees we asked to review your metanalysis.

You will see from the comments below that both referees are supportive of publication and only comment on a few items that would make the paper more understandable and comprehensive. I will not detail their comments, as they are explicit enough, but would encourage you to address all concerns.

We would welcome the submission of a revised version within three months for further consideration. Please note that EMBO Molecular Medicine strongly supports a single round of revision and that, as acceptance or rejection of the manuscript will depend on another round of review, your responses should be as complete as possible.

Please read below for important editorial formatting and consult our author's guidelines for proper

formatting of your revised article for EMBO Molecular Medicine.

I look forward to receiving your revised manuscript.

***** Reviewer's comments *****

Referee #1 (Comments on Novelty/Model System):

The type of iPSC-based disease modeling studies with low n and with poor characterization of cell types used may not provide robust data. Fewer but high quality and sufficiently powered studies could be superior and reduce noise and reporting bias

Referee #1 (Remarks):

The study by Hollingsworth et al addresses an important problem in iPSC based disease modeling in the CNS which is the difficulty of reporting, categorizing and comparing disease phenotypes across various studies, diseases and cell types based on the data published in the current literature. By selecting 93 studies (using QC criteria) the authors identify more than 600 phenotypes that are further classified into a taxonomy of 9 phenotypic clusters. The study also defines a minimal set of reporting criteria in iPSC-based disease modeling studies that is a valuable effort that could be helpful for the field. The overall goal of the study is to define novel association between disease, gene, phenotype, cell type and brain region. Such an effort could potentially reduce redundant work by establishing a database of all the disease studies and phenotypes reported for researchers to further mine. Finally, the authors establish a user-friendly dynamic online database (iPhenmap) that includes additional details that can be accessed by the reader. The online database should be updated with future studies in a timely manner and thereby could become more and more useful if properly maintained and curated.

The major challenge of the work, a challenge that the authors are well aware of, is that the power for most of the studies included is very poor and variable (remarkably low average number of 2-3 patients per study nearly all without isogenic controls!). Furthermore, there is likely a major reporting bias that could skew the data (publishing "positive" phenotypes that are expected based on other work such as for example known results from mouse studies). Despite these limitations, this is a laudable first, tour-de-force effort to address some of those challenges in the iPSC field. The current metastudy also highlights many of the current limitations and shortcomings of iPSC-based disease modeling which could provide an impetus for improving quality and statistical power in future studies.

I did like the included movie that makes it somewhat simpler to follow the work as what is presented in text and figures. On the other hand, I found the paper itself quite difficult to read as it provides very extensive descriptive data across many disorders and studies but often without a definite message of what those data mean. The text jumps forth and back from describing individual, rather random, phenotypes to trying to make broader points on iPSC phenotyping. It could benefit from a clearer outline that guides the reader in a more structured manner through the various points that authors try to make. This is also the case for some of the Figures. For several Figures, the presentation is rather complex. For example, in Figure 4A I am still unclear what the authors try to show and what the main main conclusions are (e.g. with regard to "epoch"). Similarly, Figure 3A is also complex and difficult to read and does not easily allow the reader to come to up with meaningful conclusions.

The authors further present interesting data related to cell type analyzed (for example oligodendrocytes appear to have more phenotypes in average than other neural cell types). However, there is a concern that the definition of an oligodendrocyte (as well as the definition of neurons or astrocytes) is far from standard across the studies. It may be important to discuss the need for minimal criteria to define a given cell type and maturity. For example for oligodendrocytes, the data are likely "contaminated" with by more immature NPC-type cells as true human oligodendrocytes would likely require differentiation periods of > 100 days using current differentiation technologies.

The Graphical abstract is too dense as presented with many panels and text that is too small to read.

The illustration should focus on the conceptual points to better convey the overall effort rather than trying to include too much detail within the graphical abstract.

The authors should more clearly discuss the problem that not all studies look at comparable phenotypes and therefore many associations between two genes may be due to the fact that studies on those genes looked at those phenotypes but not at other phenotypes could have similarly linked those genes. For example in PD related genes or in AD related genes, investigators will automatically steer towards certain phenotypes based on previous work which would likely skew those data across the broader gene and disease classes..

In conclusion, the study is a first effort to catalogue phenotypes from iPSC studies in a systematic manner. While I am not convinced that the current metastudy allows for any major novel conclusions at this point, I think that this is a valuable and bold effort towards this goal, and that the conclusions may sharpen in the future if the dataset continues to increase. The work clearly points to the need for studying and reporting disease phenotypes in a more standardized manner in order to fulfill the great potential of iPSC-based disease modeling.

Referee #2 (Remarks):

Hollingsworth et al conduct a meta-analysis of studies that have assessed the pathophenotypic expression in iPSCs and various neural subtypes differentiated from patients iPSCs. This study is quite comprehensive and very timely after nearly a decade of disease modeling studies using iPSCs for various neurological disorders. The authors have performed a very thorough analysis, which has culminated into iPhemap, a web resource that will be continuously updated, and that will undoubtedly be useful to gain insights into the genotype-phenotype relationship without the noise provided by the different experimental settings and designs of each individual studies. Taken together, this tool will help to further our knowledge and understanding of how, with what and for what iPSCs should be used to study disease associated phenotypes related to neurological disorders.

Minor comments:

Studies using iPSC lines derived from sporadic cases were included in the study, however there is very little mention of how the sporadic cases are being treated in the analyses performed. Wherever possible, it would be better if the distinction between mendelian forms of the disease and sporadic forms could be done in the analyses.

Related to this, in Supporting information Table 10, the disease state of the patients from which the iPSC lines are derived for all sporadic cases should be mentioned.

It would be interesting - provided that the number of studies reporting inter-clonal variations within the same lines is sufficient - to assess the difference in phenotype expression between lines derived from different clones of the same patient line.

The definition of categorical cluster descriptions could be made clearer. For instance, the difference between "impairment of expected cellular functions" and "decreased cellular processes and products" is not obvious. Furthermore, the "Absence of expected normal phenotype" cluster in supporting information Table 2, which is described as "any phenotype that can be described by the complete absence of any phenotype that is expected and found in a healthy version of the same cell" would be better described as complete loss of function.

In figure 3, the color-coding is extremely confusing between the phenotypes and the cell types. Moreover, the colors of the different phenotypes does not seem to be completely identical to that of figure 2 which adds up to the confusion.

Some cell types such as iPSCs, NSCs and astrocytes lack the presence of some of the phenotype classes (Supplemental figure 3). It should be made clearer whether these phenotype classes have been investigated but no difference was observed between the disease line(s) and controls or phenotypes belonging to these particular classes were simply not assessed in those studies.

Figure 4A: While I do like the way the results are presented in this figure, I am missing how this

illustrates the relationship between phenotypes and developmental epochs. Is the data expressed as a cell maturation timeline on the x axis i.e. are the heatmap boxes at the far left representing phenotypes observed in more mature neurons than those further right? What is the organization of the heatmap on the x axis aside from by cell types?

1st Revision - authors' response

08 September 2017

In the next several paragraphs, we will provide a point-by point answer to the reviewers' comments:

Referee #1:

We appreciate the reviewer's positive response about our manuscript, "Despite these limitations, this is a laudable first, tour-de-force effort to address some of those challenges in the iPSC field..." We have addressed the reviewer's comments as follows:

***Question 1.1** On the other hand, I found the paper itself quite difficult to read as it provides very extensive descriptive data across many disorders and studies but often without a definite message of what those data mean. The text jumps forth and back from describing individual, rather random, phenotypes to trying to make broader points on iPSC phenotyping. It could benefit from a clearer outline that guides the reader in a more structured manner through the various points that authors try to make.*

Response:

We appreciate this important reviewer comment. We have significantly restructured our manuscript to reduce the inclusion of specific observations and add more discussion about the meaning of our findings.

***Question 1.2** This is also the case for some of the Figures. For several Figures, the presentation is rather complex. For example, in Figure 4A I am still unclear what the authors try to show and what the main conclusions are (e.g. with regard to "epoch").*

Response:

We thank the reviewer for this comment and have updated **Figure 4** for increased clarity of our findings and moved the heatmap back to **Supplemental Figures**. We have also updated the figure legends and changed epoch to stage.

“Figure 4. Quantification of phenotypes by genes and developmental stage. A) Schematic diagram depicting developmental timeline of iPSC-derived cells included in analysis. B-F) Percent distribution plots of iPSC, NSC, astrocyte, oligodendrocyte, and neuronal phenotypes reported for genes linked to neurodegenerative, neurodevelopmental or other (psychiatric and viral-induced) disorders. Each data point represents a specific disease. One-way analysis of variance (ANOVA) with Bonferroni multiple comparisons tests were performed. G) Distribution of phenotypes by pluripotent, progenitor, and postmitotic cell type for Alzheimer's (AD), Parkinson's (PD), Huntington's Disease (HD), and Rett Syndrome. Two-way ANOVA with Tukey's multiple comparisons test was performed. Data are expressed as mean percentage \pm s.e.m., * $P < 0.05$, ** $P < 0.01$...”

***Question 1.3** Similarly, Figure 3A is also complex and difficult to read and does not easily allow the reader to come to up with meaningful conclusions.*

Response:

We are grateful for this reviewer comment. We agree and have added a simpler distribution plot as **Figure 3A**. We have updated the colors to be more consistent with Figure 2 to reduce any ambiguity, deleted the small numbers, highlighted the most significant associations in the Circos plot and added more description to the figure legend to guide the reader about the significance of the findings.

“Figure 3. Phenotypic classes by patient-derived cell type from 663 annotated phenotypes and phenotype: paper metric. A) Distribution of the phenotype classes within each CNS cell type with total number of phenotypes listed above each respective column. B) Circos plot of phenotype classes by cell type and vice versa are depicted by connecting ribbons, with the

width of each band proportional to the percent composition and the top-most ribbons highlighted. The neuronal ribbons (blue) were found to connect to and be largest for almost every phenotypic class. The outer track indicates the numeric percentage of phenotypic classes comprising each cell type. C) Metric of total phenotypes per cell type with respect to the total number of studies that investigated that particular cell type.”

Question 1.4 *The authors further present interesting data related to cell type analyzed (for example oligodendrocytes appear to have more phenotypes in average than other neural cell types). However, there is a concern that the definition of an oligodendrocyte (as well as the definition of neurons or astrocytes) is far from standard across the studies. It may be important to discuss the need for minimal criteria to define a given cell type and maturity. For example for oligodendrocytes, the data are likely "contaminated" with by more immature NPC-type cells as true human oligodendrocytes would likely require differentiation periods of > 100 days using current differentiation technologies.*

Response:

We acknowledge this important reviewer point. We have added discussion of the need for criterion to define a specific cell type in the MiPSCE section (**Page 19, Line 16**).

“The future inclusion of such data from differentiated cultures may help address the need for a standard set of criteria to define a given cell type, perhaps with thresholds of purity defined by marker expression and physiological measures, which would increase the reliability of comparing reported phenotypes, for it is unclear whether these differences can affect the phenotypes and gene expression of the derived cells (Figure 1).”

Question 1.5 *The Graphical abstract is too dense as presented with many panels and text that is too small to read. The illustration should focus on the conceptual points to better convey the overall effort rather than trying to include too much detail within the graphical abstract.*

Response:

We thank the reviewer for this comment and have simplified our graphical abstract to minimize distraction and guide the reader to the salient points of the work (**Page 2**).

Question 1.6 *The authors should more clearly discuss the problem that not all studies look at comparable phenotypes and therefore many associations between two genes may be due to the fact that studies on those genes looked at those phenotypes but not at other phenotypes could have similarly linked those genes. For example in PD related genes or in AD related genes, investigators will automatically steer towards certain phenotypes based on previous work which would likely skew those data across the broader gene and disease classes.*

Response:

We agree with this reviewer comment and have added discussion of this limitation (**Page 20, Line 14**).

“This potential bias may have influenced our current set of overlapping phenotypes as investigators may have been more inclined to test for phenotypes based on prior work, thereby diminishing the potential for phenotypes to link genes from different diseases. Therefore, expanding which phenotypes are tested for, outside the scope of past studies, will further enrich phenogenetic analyses.”

Referee #2:

We thank the reviewer for their positive comments and valuable suggestions to enhance the clarity and strength of our manuscript. We have addressed their comments as follows:

Question 2.1 *Studies using iPSC lines derived from sporadic cases were included in the study, however there is very little mention of how the sporadic cases are being treated in the analyses performed. Wherever possible, it would be better if the distinction between mendelian forms of the disease and sporadic forms could be done in the analyses.*

Response:

We thank the reviewer for this important point. We have clarified that only one sporadic cell line

was included in our study and the rest were disorders driven by somatic mutations. (Page 12, Line 10).

“In addition, we detected one AD-linked gene, APP, to be most concordant with an AD cell line derived from a sporadic-diseased patient with no known mutation, or “Sporadic” in Supplemental Figure 6A, the only sporadic line included in our analysis.”

Question 2.2 Related to this, in Supporting Information Table 10, the disease state of the patients from which the iPSC lines are derived for all sporadic cases should be mentioned.

Response:

We thank the reviewer for this comment and have specified the disease in **SI Table 10** as “Sporadic (AD).”

Question 2.3 It would be interesting - provided that the number of studies reporting inter-clonal variations within the same lines is sufficient - to assess the difference in phenotype expression between lines derived from different clones of the same patient line.

Response:

We acknowledge this insightful reviewer comment and agree that this analysis would be valuable, but failed to find enough papers including this information in our analysis to make it possible.

Question 2.4 The definition of categorical cluster descriptions could be made clearer. For instance, the difference between "impairment of expected cellular functions" and "decreased cellular processes and products" is not obvious. Furthermore, the "Absence of expected normal phenotype" cluster in supporting information Table 2, which is described as "any phenotype that can be described by the complete absence of any phenotype that is expected and found in a healthy version of the same cell" would be better described as complete loss of function.

Response:

We appreciate this reviewer comment and have edited the descriptions of **SI Table 2** accordingly.
Impairment of expected cellular functions: “This category contains any phenotype that can be described by the presence of a disrupted/changed state of a structure or process that is expected and found in a healthy version of the same cell and cannot be described in terms of increases or decreases. i.e. Impaired structure of adherens junctions (PMID: 24996170).”
Absence of expected normal phenotype: “This category contains any phenotype that can be described by the complete loss of a function that is found in a healthy version of the same cell. i.e. Absence of random X-inactivation (PMID: 21372419).”

Question 2.5 In figure 3, the color-coding is extremely confusing between the phenotypes and the cell types. Moreover, the colors of the different phenotypes does not seem to be completely identical to that of figure 2 which adds up to the confusion.

Response:

We thank the reviewer for this valuable comment. We agree that the colors of phenotype categories and cell types did not match precisely and have corrected this inconsistency in **Figures 2 and 3**.

Question 2.6 Some cell types such as iPSCs, NSCs and astrocytes lack the presence of some of the phenotype classes (Supplemental figure 3). It should be made clearer whether these phenotype classes have been investigated but no difference was observed between the disease line(s) and controls or phenotypes belonging to these particular classes were simply not assessed in those studies.

Response:

We acknowledge this reviewer comment and have clarified this limitation (Page 8, Line 9).
“However, given the small number of studies that modeled patient-derived glial cells, the absence of certain phenotypic clusters may be an artifact of field biases, opposed to the true state of these diseased cells.”

Question 2.7 Figure 4A: While I do like the way the results are presented in this figure, I am missing how this illustrates the relationship between phenotypes and developmental epochs. Is the

data expressed as a cell maturation timeline on the x axis i.e. are the heatmap boxes at the far left representing phenotypes observed in more mature neurons than those further right? What is the organization of the heatmap on the x axis aside from by cell types?

Response:

We thank the reviewer for pointing out this ambiguity and have updated **Figure 4A** for clarity, moved the heatmap back to **Supplemental Figures**, and replaced epoch with stage. We also have updated both of their legends to guide the reviewer through their findings.

“Figure 4. Quantification of phenotypes by genes and developmental stage. A) Schematic diagram depicting developmental timeline of iPSC-derived cells included in analysis. B-F) Percent distribution plots of iPSC, NSC, astrocyte, oligodendrocyte, and neuronal phenotypes reported for genes linked to neurodegenerative, neurodevelopmental or other (psychiatric and viral-induced) disorders. Each data point represents a specific disease. One-way analysis of variance (ANOVA) with Bonferroni multiple comparisons tests were performed. G) Distribution of phenotypes by pluripotent, progenitor, and postmitotic cell type for Alzheimer’s (AD), Parkinson’s (PD), Huntington’s Disease (HD), and Rett Syndrome. Two-way ANOVA with Tukey’s multiple comparisons test was performed. Data are expressed as mean percentage \pm s.e.m., * $P<0.05$, ** $P<0.01$...”

We would like to thank the reviewers and editors. Your recommendations certainly have made our contribution more robust to the field of iPSCs as an “eye-opening” work.

Corresponding Author Name: Jaime Imitola

Manuscript Number: EMM-2017-08191